# Bi-objective Trade-off with Dynamic Barrier Gradient Descent

**Chengyue Gong**      **Xingchao Liu**      **Qiang Liu**

Computer Science, The University of Texas at Austin

{cygong17,xcliu,lqiang}@cs.utexas.edu

## Abstract

Many machine learning tasks have to make a trade-off between two loss functions, typically the main data-fitness loss and an auxiliary loss. The most widely used approach is to optimize the linear combination of the objectives, which, however, requires manual tuning of the combination coefficient and is theoretically unsuitable for non-convex functions. In this work, we consider constrained optimization as a more principled approach for trading off two losses, with a special emphasis on *lexicographic (lexico) optimization*, a degenerated limit of constrained optimization which optimizes a secondary loss inside the optimal set of the main loss. We propose a *dynamic barrier gradient descent* algorithm which provides a unified solution of both constrained and lexicographic optimization. We establish the convergence of the method for general non-convex functions. Through a number of experiments on real-world deep learning tasks, we show that 1) lexico optimization provides a tuning-free approach to incorporating side loss functions without hurting the main objective, and 2) constrained and lexico optimization combined provide an automatic approach to profiling Pareto sets, especially in non-convex problems on which linear combination methods fail.

## 1   Introduction

Although machine learning (ML) has been typically conceptualized as optimizing a single objective function, most practical ML tasks actually involve trading off two or more objective functions, such as the data fitness function vs. a regularization or auxiliary loss. Let $f$ and $g$ be two objectives functions of interest on $\mathbb{R}^d$. A principled way to trade-off $f$ and $g$ is through constrained optimization:

$$\min_{\theta \in \mathbb{R}^d} f(\theta) \quad s.t. \quad g(\theta) \leq c, \tag{1}$$

where $c$ is a threshold. When varying $c$, solutions of (1) cover the Pareto points of $(f, g)$, providing different trade-offs on $f, g$. Moreover, (1) naturally arises when $g$ is an important constraint that should be controlled explicitly, e.g., in terms of safety, fairness, and other trustworthy measures.

However, compared to unconstrained optimization, constrained optimization (1) has been much less widely used in practical machine learning. In fact, perhaps the most common approach to handling (1) is to transform it into the unconstrained optimization of the *linear combination* of $f$ and $g$,

$$\min_{\theta \in \mathbb{R}^d} f(\theta) + \lambda g(\theta), \tag{2}$$

where the trade-off between $f$ and $g$ is controlled by a weight coefficient $\lambda$, instead of the threshold parameter $c$. A folklore argument is that $\lambda$ can be viewed as the Lagrange multiplier of (1), and hence (1) can be mapped into (2) with properly selected $\lambda$.

Unfortunately, despite being broadly used, (2) is insufficient to fully replace (1) and suffers from a number of disadvantages in both theory and practice:

35th Conference on Neural Information Processing Systems (NeurIPS 2021).

- *Interpretability.* $\lambda$ provides a less intuitive parameter to select than $c$, since it depends on the relative scale of $f$ and $g$, whose range is problem-dependent and needs to be optimized as a hyper-parameter. In comparison, the threshold $c$ can be specified as a tolerance parameter when $g$ is a metric that users want to control explicitly.

- *Pareto Coverage.* When $f$, $g$ are non-convex functions, (1) provides a strictly broader class of problems than (2), because for some $c$, there may exist no $\lambda \in \mathbb{R}$, such that (1) and (2) are equivalent. From the multi-objective optimization (MOO) perspective, this is related to the fact that (2) can only capture the convex envelop of the Pareto front while varying $\lambda$, while (1) provides all Pareto optimal points by varying $c$.

- *Invariance.* The constrained optimization (1) is invariant to arbitrary monotonically increasing maps on $f$ and $g$. That is, let $\psi$ and $\psi'$ be two monotonically increasing maps, then we obtain the equivalent problem if we replace $f$ with $\psi \circ f$, and $g$ with $\psi' \circ g$, and $c$ with $\psi'(c)$. In comparison, applying nonlinear transforms on $f$ and $g$ in (2) yields fundamentally different problems.

- *Harmless Regularization.* In many practical cases, one of the objectives (say $f$) is of secondary importance w.r.t. the other one (say $g$), in sense that we are interested in minimizing $f$ only when $g$ is fully optimized; this can be formulated as the following *lexicographic optimization* problem:

$$\min_{\theta \in \mathbb{R}^d} f(x) \quad s.t. \quad g(\theta) \leq g^* \qquad \text{where} \qquad g^* := \inf_{\theta \in \mathbb{R}^d} g(\theta), \qquad (3)$$

where we minimize $f$ inside the optimum set $\{\theta \colon g(\theta) \leq g^*\}$ of $g$. This can be viewed as a special case of (1) with the minimum threshold $c = g^*$ (which is unknown before hand), and can not be captured by (2) with a finite and fixed $\lambda$. Compared with constrained optimization (1), the lexicographic (or simply lexico) problem eliminates the need of setting the threshold $c$, and hence provides a handy, coefficient-free approach for incorporating auxiliary losses without hurting the main loss. As our empirical results show, this finds useful in numerous deep learning tasks, e.g., fairness ML and semi-supervised learning.

**Our Contributions** We conduct a joint study on the constrained and lexico optimization problems. We propose a simple and general local descent algorithm (Algorithm 1) which offers a unified solution to both problems. By placing a *dynamic barrier* constraint on the search direction at each iteration, the algorithm finds a trajectory towards the optimal solution by properly balancing $f$ and $g$ with an adaptive combination coefficient coefficient $\lambda_t$; as seen in Eq 4, the $\lambda_t$ is decided with a simple formula by the inner prod uct between the objective and constraint gradients $\nabla f$ and $\nabla g$.

In Section 3.1-3.3, we study the continuous-time convergence of the method for general non-convex functions and both the constrained and lexico optimization cases. In Section 3.4, we elaborate the (often overlooked) fact that methods based on optimizing the linear combination (2) is fundamentally unsound as an approach to constrained optimization (1) with non-convex functions and non-zero duality gap. In Section 4, we empirically show that our method provides an efficient approach to approximating Pareto sets and incorporating side information in a variety of deep learning tasks.

---

**Algorithm 1** Dynamic Barrier Gradient Descent for (1) and (3)

---

Choose stepsize $\{\epsilon_t\}$ and the dynamic barrier function $\phi$ in (8) (use $\alpha = \beta = 1$ by default; set $\hat{g} = c$ for constrained optimization (1) and $\hat{g}$ to be any value no larger than $g^*$ for lexicographic optimization (3)).

**for** iteration $t$ **do**

$$\theta_{t+1} \leftarrow \theta_t - \epsilon_t(\nabla f(\theta_t) + \lambda_t \nabla g(\theta_t)), \quad \lambda_t = \max\left(\frac{\phi(\theta_t) - \nabla f(\theta_t)^\top \nabla g(\theta_t)}{\|\nabla g(\theta_t)\|^2}, \quad 0\right). \quad (4)$$

**end for**

---

## 2 Main Method

We introduce the main algorithm 1, a simple gradient-based method for solving both constrained optimization (1) and lexicographic optimization (3) in a unified way. The method performs iterative updates of form

$$\theta_{t+1} \leftarrow \theta_t - \epsilon_t v_t, \qquad (5)$$

where $\epsilon_t \geq 0$ is a step size and $v_t \in \mathbb{R}^d$ is an update direction to be chosen to balance the minimization of $f$ and constraint satisfaction on $g$. The $v_t$ is designed to satisfy the following desiderata:

1) When the constraint is not satisfied (i.e., $g(\theta_t) > c$), we should mainly focus on decreasing $g$ to meet the constraint as fast as possible; meanwhile, $f$ should act as a *secondary objective* in this phase, meaning that $f$ should be minimized upto the degree that it does not hurt the descent of $g$.

2) When the constraint is met (i.e., $g(\theta_t) \leq c$), we should prioritize to minimize $f$, which is made possible in general only if we allow $g$ to increase. However, the increasing rate of $g$ should be properly controlled, so that $\theta$ stays inside or nearby the feasible set while we minimize $f$.

It turns out that both properties can be achieved if we select $v_t$ by the following optimization:

$$v_t = \arg\min_{v \in \mathbb{R}^d} \left\{ \|\nabla f(\theta_t) - v\|^2 \quad s.t. \quad \nabla g(\theta_t)^\top v \geq \phi(\theta_t) \right\}, \tag{6}$$

where we want $v_t$ to be as close to $\nabla f(\theta_t)$ as much as possible (and hence decrease $f$), but subject to a lower bound on the inner product of $\nabla g(\theta_t)$ and $v_t$ to ensure that the change of $g$ is properly controlled by the location of $\theta_t$; here the lower bound $\phi \colon \mathbb{R}^d \to \mathbb{R}$ is a *dynamic barrier* function which trade-offs loss minimization with constraint satisfaction by controlling the inner product between $\nabla g(\theta_t)$ and $v_t$. To achieve the desiderata on $v_t$, we should let $\phi(\theta_t)$ have the same sign as $g(\theta_t) - c$, so that the constraint $\{\theta \colon g(\theta) \leq c\}$ is equivalent to $\{\theta \colon \phi(\theta) \leq 0\}$, that is,

$$\text{sign}(\phi(\theta)) = \text{sign}(g(\theta) - c), \tag{7}$$

where $\text{sign}(x) = \frac{x}{|x|}$ for $x \neq 0$ and $\text{sign}(0) = 0$. In this way, when the step size $\epsilon_t$ is sufficiently small, we have the following properties that will be studied theoretically in Section 3:

1) When $\theta_t$ is outside of the feasible set ($g(\theta_t) > c$), the constraint is $\nabla g(\theta_t)^\top v_t \geq \phi(\theta_t) > 0$, which ensures that $g$ decreases strictly outside of the feasible set.

2) When $\theta_t$ is on the boundary ($g(\theta_t) = c$), the constrain reduces to $\nabla g(\theta_t)^\top v_t \geq \phi(\theta_t) = 0$, meaning that we want to decrease $f$ subject to that $g$ does not increase.

3) When $\theta_t$ is in the interior of the feasible set ($g(\theta_t) < c$), the lower bound on $\nabla g(\theta_t)^\top v_t$ is negative, allowing $g$ to increase. In this case, the optimal solution $v_t$ in (6) can be shown to have a positive inner product with $\nabla f(\theta_t)$ unless $\nabla f(\theta_t) = 0$, which means that the algorithm monotonically decreases $f$ until a local optimum is reached.

**Computing $\lambda_t$**  Since (6) is a simple quadratic convex programming, it is easy to see that the solution is $v_t = \nabla f(\theta_t) + \lambda_t \nabla g(\theta_t)$ where $\lambda_t$ solves the following dual problem of (6):

$$\lambda_t = \arg\min_{\lambda \geq 0} \left\{ \|\nabla f(\theta_t) + \lambda \nabla g(\theta_t)\|^2 - \lambda \phi(\theta_t) \right\} = \max\left( \frac{\phi(\theta_t) - \nabla f(\theta_t)^\top \nabla g(\theta_t)}{\|\nabla g(\theta_t)\|^2}, \quad 0 \right).$$

**Practical Choice of $\phi$**  There are a broad range of $\phi$'s that satisfy (7). Two particularly simple choices stand out which are suitable for constrained and lexico optimization, respectively:

1) $\phi(\theta_t) = \alpha(g(\theta_t) - c)$, where $\alpha > 0$. This puts a strong requirement on descending the constraint $g$ when $g(\theta_t) - c$ is large and positive, and it gives higher flexibility for minimizing $f$ when $g(\theta_t) - c$ is negative as we move towards the interior of the feasible set. In this case, the inner product constraint in (6) can be viewed as a linearization of the original constraint ($g(\theta) \leq c$) in (1): $g(\theta_t - \epsilon_t v_t) \approx g(\theta_t) - \epsilon_t \nabla g(\theta_t)^\top v_t \leq c$, with $\epsilon_t = 1/\alpha$. Hence, the algorithm in this case can be viewed as a simple variant of sequential quadratic programming (SQP) (e.g., Nocedal & Wright, 2006) when we use an identity matrix to approximate the Hessian of $f$.

2) $\phi(\theta_t) = \beta \|\nabla g(\theta_t)\|^2$, where $\beta > 0$. Assume that $\nabla g(\theta_t) = 0$ implies global minimum of $g$, then this $\phi$ satisfies the sign condition (7) for the lexico case (3), without requiring to estimate the unknown threshold $g^*$. Because $\phi(\theta_t) \geq 0$ in this case, it allows us to strictly decrease $g$ until $\theta_t$ reaches an optimum of $g$. Meanwhile, $f$ acts as the secondary objective that is decreased when it does not interfere with the descent of $g$. In practice, when $g$ has multiple local minima, then the algorithm would minimize $f(\theta)$ subject to $g(\theta) \leq g(\hat\theta^*)$, where $\hat\theta^*$ is a local minimum of $g$ that the algorithm encounters.

Combining the two cases into a single $\phi$, we propose to use

$$\phi(\theta) = \min\left(\alpha(g(\theta) - \hat{g}), \ \beta \|\nabla g(\theta)\|^2\right), \tag{8}$$

where we take $\hat{g} = c$ if the goal is to solve the constrained optimization (1) with threshold $c$, and take $\hat{g}$ to be any lower bound of $g^*$ (i.e., $\hat{g} \leq g^*$) for solving the lexicographic optimization problem (3); for example, if $g$ is a norm or mean square loss, then $\hat{g} = 0$ is a natural estimation of the lower bound. If no lower bound can be estimated, we can take $\hat{g} = -\infty$, and (8) reduces back to $\phi(\theta) = \beta \|\nabla g(\theta)\|^2$. The choice of the hyper-parameters $\alpha, \beta$ controls the speed of constraint satisfaction vs. loss minimization. In our experiments, we find that it is sufficient to simply set $\alpha = \beta = 1$ in most cases so that the algorithm introduces no extra hyper-parameters compared with vanilla gradient descent. See Algorithm 1 for a summary of the algorithm procedure.

**Profiling Pareto Set**   The Pareto set of $(f, g)$ is the set of points $\theta$ for which there exists no $\theta'$, such that $f(\theta') \leq f(\theta)$, $g(\theta') \leq g(\theta)$, and $(f(\theta'), g(\theta')) \neq (f(\theta), g(\theta))$. The combination of the constrained and lexico optimization provides an automatic approach to obtain uniformly distributed points from the Pareto set of $(f, g)$. To do so, we start with solving the lexico problem (3), which corresponds to one of the endpoints of the Pareto set; we then walk through the Pareto points by solving the constrained optimization (1) with $c$ increasing linearly from the $g^*$ estimated from lexico optimization. Optionally, we can also solve the mirror version $\min_\theta g(\theta) \ s.t. \ f(\theta) \leq f^* := \inf_{\theta'} f(\theta')$ to obtain the other end point, and then solve (1) with $c$ on a grid between the two end points.

## 3   Theoretical Analysis

We analyze the convergence of Algorithm 1 for solving both constrained optimization (Section 3.1) and lexico optimization (Section 3.3). A key of our results is that they hold for general non-convex functions, which is essential for deep learning applications. In comparison, we show in Section 3.4 that the commonly used methods that transform constrained optimization (1) into a sequence of unconstrained optimization problems (2) rely on the strong duality assumption and are hence unsound for non-convex cases. For simplicity, we focus on the continuous-time limit of the algorithm with $d\theta_t/dt = -v_t$, where $t \in [0, +\infty)$ denotes the continuous time.

### 3.1   Basic Characterization: KKT Score and $L_1$ Penalty Function

We provide a basic characterization of our method in connection with the first-order KKT condition and the $L_1$ penalty function. We start with recalling the first-order KKT necessary condition (Nocedal & Wright, 2006) of the constrained optimization (1): Assume $\theta^*$ is a local optimal solution of (1), that $f$ and $g$ are continuously differentiable, and that $\|\nabla g(\theta^*)\| \neq 0$, then there exists a Lagrange multiplier $\lambda^* \in [0, +\infty)$, such that

*stationarity:* $\nabla f(\theta^*) + \lambda^* \nabla g(\theta^*) = 0$,   *Feasibility:* $g(\theta^*) \leq c$,   *Slackness:* $\lambda^*(g(\theta^*) - c) = 0$.

Note that $\|\nabla g(\theta^*)\| \neq 0$ is an important regularity condition known as a *constraint quantification condition*. In particular, in the lexico case ($c = g^*$), we always have $\|\nabla g(\theta^*)\| = 0$, and hence the KKT condition above does not hold (we may think that $\lambda^* = +\infty$ in this case). However, the results in this subsection (Theorem 3.2 and Collorary 3.3) remain to be true for the lexico case. We discuss a different second-order KKT condition for lexico case in Section 3.3.

Assume that $\phi$ satisfies the sign condition (7). For $\lambda \geq 0$, we can use the following score function, which we call the *KKT score*, to assess the KKT condition:

$$K_\nu(\theta, \lambda) = \|\nabla f(\theta) + \lambda \nabla g(\theta)\|^2 + \nu[\phi(\theta)]_+ + \lambda[-\phi(\theta)]_+, \tag{9}$$

where $\nu > 0$ is a coefficient and $[x]_+ = \max(x, 0)$; it is easy to see that $K_\nu(\theta, \lambda) \geq 0$ for $\theta \in \mathbb{R}^d$ and $\lambda \geq 0$, and for $(\theta, \lambda) \in \mathbb{R}^d \times [0, \infty)$, we have $K_\nu(\theta, \lambda) = 0$ iff $(\theta, \lambda)$ meets the KKT condition.

A different way to access the optimality of (1) is to use the following $L_1$ penalty function:

$$P_\mu(\theta) = f(\theta) + \mu[g(\theta) - c]_+, \tag{10}$$

where $\mu > 0$ is a coefficient. It is known that the minimum points of (10) coincide with the solution of (1) if $\mu$ is taken to be sufficiently large (see e.g., Nocedal & Wright (2006)).

Below is a basic property of Algorithm 1 in terms of the KKT score $K_\nu(\theta, \lambda)$ and the penalty function $P_\mu(\theta)$, which underpins all subsequent results regarding the algorithm.

**Assumption 3.1.** *Assume $f$ and $g$ are continuously differentiable. Let $\{\theta_t \colon t \in [0, +\infty)\}$ be governed by the continuous-time dynamics $d\theta_t/dt = -v_t$ with $v_t$ defined in (6) and $\phi$ satisfying the sign condition (7). Assume $\lambda_t < +\infty$ for $t \in [0, +\infty)$.*

**Theorem 3.2.** *Assume Assumption 3.1 holds. We have for any $\mu \geq 0$,*

$$\frac{d}{dt} P_\mu(\theta_t) \leq -K_{\mu - \lambda_t}(\theta_t, \lambda_t), \quad \forall t \in [0, \infty). \tag{11}$$

Eq (11) shows that, at any time $t$, the penalty function $P_\mu(\theta_t)$ is non-increasing w.r.t. time $t$ if $K_{\mu - \lambda_t}(\theta_t, \lambda_t) \geq 0$, which, according to the definition of KKT score in (9), holds if either $\mu$ is large enough so that $\mu - \lambda_t \geq 0$, or the constraint is met (i.e., $g(\theta_t) \leq c$ and hence $[\phi(\theta_t)]_+ = 0$).

Taking different values of $\mu$ allows us to extract some basic properties of the algorithm. By taking $\mu \to +\infty$ in (11), we show below that the constraint $[g(\theta_t) - c]_+$ is always non-increasing w.r.t. $t$, meaning that $g(\theta_t)$ is decreasing w.r.t. $t$ outside of the feasible region and $\theta_t$ stays within the feasible region once it is reached for the first time. On the other hand, by taking $\mu = 0$ in (11), we can show that $f(\theta_t)$ decreases monotonically w.r.t. time $t$ within the feasible set, until a KKT point is found.

**Corollary 3.3.** *Assume Assumption 3.1 holds. We have:*

*1) The thresholded constraint function $[g(\theta_t) - c]_+$ is always non-increasing w.r.t. time $t$:*

$$\frac{d}{dt}[g(\theta_t) - c]_+ \leq -[\phi(\theta_t)]_+ \leq 0, \qquad \forall t \in [0, \infty), \tag{12}$$

*which yields that $\int_0^t [\phi(\theta_s)]_+ ds \leq [g(\theta_0) - c]_+$, and hence $\min_{s \in [0,t]} [\phi(\theta_s)]_+ = O(1/t)$.*

*2) If the constraint is satisfied (i.e., $g(\theta_t) \leq c$), the objective $f(\theta_t)$ is non-increasing w.r.t. time $t$:*

$$\frac{d}{dt} f(\theta_t) \leq -\|\nabla f(\theta_t) + \lambda_t g(\theta_t)\|^2 - \lambda_t[-\phi(\theta_t)]_+ \leq 0, \qquad when \qquad g(\theta_t) \leq c. \tag{13}$$

### 3.2 Constrained Optimization with Bounded $\lambda_t$

For constrained optimization (1) with a finite Lagrange multiplier $\lambda^*$, it is natural to expect that $\lambda_t$ is bounded by a finite number $\lambda_{\max}$ outside of the feasible set $\{\theta \colon g(\theta) \leq c\}$. In this case, it is easy to see from (11) that any penalty function $P_\mu(\theta)$ with $\mu \geq \lambda_{\max}$ is a Lyapunov function of the algorithm, and the best KKT score upto time $t$ (i.e., $\min_{s \in [0,t]} K_\nu(\theta_s, \lambda_s)$) decays with $O(1/t)$ rate for any $\nu \geq 0$.

**Assumption 3.4.** *Assume $\sup_t \{\lambda_t \colon g(\theta_t) > c, \quad t \in [0, \infty)\} = \lambda_{\max} < \infty$.*

**Corollary 3.5.** *Under Assumption 3.1 and 3.4, we have:*

*1) The penalty function $P_\mu(\theta_t)$ with $\mu \geq \lambda_{\max}$ is non-increasing w.r.t. time $t$ :*

$$\frac{d}{dt} P_\mu(\theta_t) \leq -K_{\mu - \lambda_{\max}}(\theta_t, \lambda_t) \leq 0, \qquad \forall t \in [0, +\infty). \tag{14}$$

*2) Assume $f^* = \inf_{\theta \in \mathbb{R}^d} f(\theta) > -\infty$. The KKT score $K_\nu(\theta_t, \lambda_t)$ with any $\nu \geq 0$ satisfies*

$$\int_0^t K_\nu(\theta_s, \lambda_s) ds \leq (P_{\nu + \lambda_{\max}}(\theta_0) - f^*), \qquad \forall t \in [0, +\infty),$$

*which yields that $\min_{s \in [0,t]} K_\nu(\theta_s, \lambda_s) = O(1/t)$.*

*3) If $\theta_t$ is a fixed point in the sense that $v_t = 0$, with $\lambda_t < \infty$, then $(\theta_t, \lambda_t)$ satisfies the first-order KKT necessary condition above, i.e., $K_\nu(\theta_t, \lambda_t) = 0$ for any $\nu \geq 0$.*

Below we provide a condition when Assumption 3.4 of finite $\lambda_{\max}$ holds.

**Proposition 3.6.** *Assume Assumption 3.1 holds. Let $c_0 = \max(c, g(\theta_0))$. Then, we have:*

*1) The trajectory $\{\theta_t \colon t \in [0, \infty)\}$ is contained inside the sublevel set $\{\theta \colon g(\theta) \leq c_0\}$.*

*2) If $\max(\phi(\theta), \|\nabla f(\theta)\|, 1/\|\nabla g(\theta)\|) < \infty$ for $\forall \theta \in \Gamma := \{\theta \in \mathbb{R}^d \colon c < g(\theta) \leq c_0\}$, then Assumption 3.4 holds (i.e., $\lambda_{\max} < \infty$).*

The most significant assumption in Proposition 3.6 is that $1/\|\nabla g(\theta)\| < \infty$ for $\theta \in \Gamma$. If $\Gamma$ is compact and $\nabla g(\theta)$ is continuous, it can be replaced by a weaker condition of $\|\nabla g(\theta)\| \neq 0$, $\forall \theta \in \Gamma$. This condition is necessary because otherwise there can be local optima of $g$ outside of the feasible set, in which case a local descent algorithm may get stuck at an infeasible local optimum and hence can not guarantee to reach the feasible set. This condition can be replaced by a weaker one as we discuss in the proof of Proposition 3.6.

### 3.3 Lexicographic Optimization with Unbounded $\lambda_t$

When applying Algorithm (1) to the lexicographic optimization 3, we expect that $\|\nabla g(\theta_t)\|$ decays to zero and hence $\lambda_t$ converges to $+\infty$. Correspondingly, the KKT condition in Section 3.1 does not hold since we would have $\lambda^* = +\infty$. However, it is important to point out that the KKT score $K_\nu(\theta, \lambda)$ still indicates useful information regarding local optimality even in the lexico case. To see this, note that in the lexico case, we always have $\phi(\theta) \geq 0$ and hence

$$K_\nu(\theta_t, \lambda_t) = \|\nabla f(\theta_t) + \lambda_t \nabla g(\theta_t)\|^2 + \nu \phi(\theta_t),$$

where $\phi(\theta_t) = 0$ indicates that $\theta_t$ is a stationary (and hopefully minimum) point of $g$, and $\|\nabla f(\theta_t) + \lambda_t \nabla g(\theta_t)\| = \|v_t\| = 0$ indicates the local optimality w.r.t. $f$, because it is the KKT condition of a relax problem $\min_\theta \{f(\theta) \ s.t. \ g(\theta) \leq c_t\}$, where $c_t = g(\theta_t)$.

In the following, we show that with a proper choice of $\phi$ in the lexico case, the algorithm still decays the best KKT score $\min_{s \in [0,t]} K_\nu(\theta_s, \lambda_s)$ upto time $t$, despite with a rate slower than $O(1/t)$.

**Proposition 3.7.** *Assume Assumption 3.1 holds, and $c = g^* = \inf_{\theta \in \mathbb{R}^d} g(\theta) > -\infty$, and $f^* = \inf_{\theta \in \mathbb{R}^d} f(\theta) > -\infty$. Further, assume $\phi$ satisfies*

$$0 \leq \phi(\theta_t) \leq \beta \|\nabla g(\theta_t)\|^\tau, \qquad where \ \beta > 0 \ and \ \tau \geq 1.$$

*Then we have for any time $t \in [0, \infty)$,*

$$\min_{s \in [0,t]} \phi(\theta_s) \leq \frac{\Delta_g}{t}, \qquad \min_{s \in [0,t]} \|v_s\|^2 \leq C_0 \left(\frac{\Delta_g}{t}\right)^{1 - \frac{1}{\tau}} + \frac{\Delta_f}{t},$$

*where $\Delta_g = g(\theta_0) - g^*$, $\Delta_f = f(\theta_0) - f^*$, and $C_0 = \sup_{\theta \in \mathbb{R}^d} \left(\beta \|\nabla g(\theta)\|^{\tau - 1} + \|\nabla f(\theta)\|\right) \beta^{\frac{1}{\tau}}$.*

This suggests that $\min_{s \in [0,t]} K_\nu(\theta_s, \lambda_s) = O(1/t^{1 - 1/\tau})$ for a fixed $\nu \geq 0$. If we take $\phi(\theta) = \beta \|\nabla g(\theta)\|^\tau$, we have $\min_{s \in [0,t]} \|\nabla g(\theta_s)\|^2 = O(1/t^{2/\tau})$ and $\min_{s \in [0,t]} \|v_s\|^2 = O(1/t^{1 - 1/\tau})$. Therefore, the power index $\tau$ controls the relative convergence speed of $\|\nabla g(\theta_t)\|$ (measuring the minimization of $g$), and that of $\|v_t\|$ (measuring the minimization of $f$).

**The Lexico KKT Condition** Although the first order KKT condition in Section 3.1 does not apply, a different KKT condition that involves the Hessian matrix $\nabla^2 g$ can be derived for lexico optimization (see e.g., Dempe et al. (2010)). To see this, we relax the lexico problem (3) into

$$\min_{\theta \in \mathbb{R}^d} f(\theta) \quad s.t. \quad \nabla g(\theta) = 0. \tag{15}$$

If $\theta^*$ is a local optimum of (3), then it is also a local optimum of (15). Assume $f$ and $\nabla g$ are continuously differenitable, and the rank of the Hessian matrix $\nabla^2 g(\theta)$ equals a constant in a neighborhood of $\theta^*$ (known as the *constant rank constraint quantification* (Janin, 1984)). Then the first-order KKT necessary condition of (15) says that, there exists a vector-valued Lagrange multiplier $\omega^* \in \mathbb{R}^d$, such that

$$\nabla f(\theta^*) + \nabla^2 g(\theta^*) \omega^* = 0. \tag{16}$$

This is equivalent to that $\nabla f(\theta^*)$ is orthogonal to the null space of $\nabla^2 g(\theta^*)$, which is the tangent space of stationary manifold $\{\theta \colon \nabla g(\theta) = 0\}$ of $g$.

In Proposition 6.2 in Appendix , we show that a point $\theta^*$ satisfies the lexico KKT condition in (16) if it is a limit of a sequence $\{\theta_t\}$ that satisfies $\lim_{t \to +\infty} K_\nu(\theta_t, \lambda_t) = 0$ for a positive $\nu$. With this, if our algorithm converges to a point $\theta^*$, it satisfies the lexico KKT condition.

**Proposition 3.8.** *Assume that the conditions in Proposition 3.7 holds, and $\phi$ is continuous w.r.t. $\theta$. Assume $\nabla g$ is continuously differentiable. If $\theta_t$ converges to a limit point $\theta^*$ with $\lim_{t \to \infty} \theta_t = \theta^*$ and $\lim_{t \to \infty} v_t = 0$, and the rank of the Hessian matrix $\nabla^2 g(\theta)$ equals a constant in a neighborhood of $\theta^*$, then there exists $\omega^* \in \mathbb{R}^d$ such that $\theta^*$ satisfies the lexico KKT condition in (16).*

## 3.4 Non-convexity and Primal vs. Dual Methods

Many popular approaches for constrained optimization (1) used in machine learning are based on transforming it into a sequence of unconstrained optimization of form (2) with $\lambda$ iteratively updated in certain way by viewing it as the Lagrange multiplier. However, all these methods based on (2) are fundamentally *dual methods* which are guaranteed to work only when the strong duality holds. In comparison, Algorithm 1 is a *primal method*, which directly solves the primal problem and is hence more suitable for non-convex functions.

To elaborate, note that the constrained optimization is equivalent to the following minimax problem:

$$\min_{\theta \in \mathbb{R}^d} \max_{\lambda \geq 0} f(\theta) + \lambda(g(\theta) - c). \tag{17}$$

If the strong duality holds, exchanging the order of min and max yields an equivalent dual problem:

$$\max_{\lambda \geq 0} \Phi(\lambda), \qquad \Phi(\lambda) = \min_{\theta \in \mathbb{R}^d} \{f(\theta) + \lambda(g(\theta) - c)\}. \tag{18}$$

Therefore, one can estimate $\lambda^*$ by maximizing $\Phi(\lambda)$ with gradient ascent, yielding the *dual ascent* algorithm. Because calculating $\Phi(\lambda)$ requires to solve the whole unconstrained optimization (2) and is costly, a simplification is to alternate between the gradient ascent on $\lambda$ and gradient descent on $\theta$ for solving the minimax problem, yielding the following *primal-dual gradient method*:

$$\lambda_t \leftarrow [\lambda_{t-1} + \xi(g(\theta_t) - c)]_+, \qquad \theta_{t+1} \leftarrow \theta_t - \epsilon(\nabla f(\theta_t) + \lambda_t \nabla g(\theta_t)), \tag{19}$$

where $\epsilon$ and $\xi$ are step sizes.

Unfortunately, all these methods, which *decouple* the estimation of $\theta$ and $\lambda$, are not suitable for non-convex functions that lack strong duality. In fact, when $f$ and/or $g$ are sufficiently non-convex, *the true solution $\theta^*$ can be a local maximum (rather than local minimum), or saddle point of $f(\theta) + \lambda^* g(\theta)$*, which happens once $\nabla^2 f(\theta^*) + \lambda^* \nabla^2 g(\theta^*)$ is not positive (semi-)definite, where $\lambda^*$ is the Lagrangian multiplier associated with $\theta^*$ in the KKT condition. In this case, even if we know the true Lagrange multiplier $\lambda^*$, minimizing $f(\theta) + \lambda^* g(\theta)$ as advocated in (2) would not yield the true solution $\theta^*$. Similarly, the primal-dual update in (19) would fail in this case because one can show that $(\theta^*, \lambda^*)$ is an unstable fixed point of (19). The augmented Lagrange methods (Bertsekas, 2014), which amount to replacing the objective $f$ with $f_\rho(\theta) := f(\theta) + \rho[g(\theta) - c]_+^2, \rho > 0$, can help "convexify" the objective if $[g(\theta) - c]_+^2$ is convex (such as the case of linear constraints), but may not help when $[g - c]_+^2$ is non-convex. In comparison, Algorithm 1 works for general non-convex functions we show in Section 3.1-3.3.

**Example 3.9.** *Consider a toy problem:* $\min_{\theta \in \mathbb{R}} |\theta - 1|^\alpha$ *s.t.* $|\theta|^\alpha \leq b^\alpha$ *with* $b \in (0, 1)$ *and* $\alpha > 0$. *Note that all* $\alpha > 0$ *yields an equivalent problem whose true solution is* $\theta^* = b$ *and Lagrange multiplier* $\lambda^* = (1 - b)^{\alpha - 1}/b^{\alpha - 1}$. *However, the choice of* $\alpha$ *changes the convexity of the problem. One can show that if* $\alpha < 1$, *the true solution is a local maximum of* $f(\theta) + \lambda^* g(\theta)$, *and hence methods based on minimizing* $f(\theta) + \lambda^* g(\theta)$ *would fail. However, Algorithm 1 solves the problem correctly for all* $\alpha > 0$. *See Appendix for more analysis.*

## 3.5 Related Works

**Constrained Optimization** There is a large classical literature on constrained optimization (Nocedal & Wright, 2006; Bertsekas, 2014; Bonnans et al., 2006), including the primal methods such as the penalty function methods, interior point methods, feasible direction methods, sequential quadratic programming (SQP), and Lagrange multiplier and dual methods. Our method is mostly related to SQP (Nocedal & Wright, 2006), of which Algorithm 1 can be viewed as a special case if we take $\phi = \alpha(g - c)$. However, the general dynamic barrier view allows a broader class of $\phi$, which is necessary for the lexico case; importantly, Algorithm 1 is fast and simple to implement, making it suitable for large-scale deep learning tasks, which is the main target domain of the method. In comparison, the traditional SQP methods tend to be complicated/expansive and have not found wide applications in large-scale deep learning tasks. The penalty function methods are widely used but require a use of large penalty coefficient which may cause numerical issues. There is a vast literature on Lagrange multiplier and (primal-)dual methods (Bertsekas, 2014), which, however, are designed for convex problems and problematic for non-convex problems as we show in Section 3.4. The high-level idea of dynamic barriers is similar to the control barrier functions in control theory (e.g., Ames et al., 2019), which does not consider solving the constraint optimization and its theoretical connection to KKT conditions.

**Lexicographic Optimization**   Despite offering a promising tool for incorporating secondary loss functions, lexico optimization (3) has been still under-explored in machine learning. In the bi-level optimization literature, (3) is known as simple bi-level programming (SBP), which is a special case of optimization on efficient sets, and more generally mathematical program with equilibrium constraints (MPEC) (Dempe et al., 2020, 2010). A number of algorithms (e.g., Sabach & Shtern (2017); Solodov (2007b,a)) have been proposed in the optimization literature, which, however, focus on convex functions and yield double loop algorithms that repeatedly call an inner loop optimization and are more expensive than our method. Kissel et al. (2020) proposes a method of lexicographic optimization for training neural networks by descending in the null space of batch activations, which, however, is not scalable to large datasets and models.

**Approximating Pareto Set**   The method of constructing Pareto sets using constrained optimization is known as the $\varepsilon$-constrained method in the multi-objective optimization literature (e.g., (Mavrotas, 2009)), which improves over the linear scalarization approach (2) for non-convex Pareto sets. However, the traditional $\varepsilon$-constrained method does not consider the lexico case and hence needs extra steps/information to decide the range of the threshold $c$. There is a vast literature of other approaches that work for non-convex Pareto sets in the MOO literature (Miettinen, 2012; Hwang & Masud, 2012; Pardalos et al., 2017), which has different advantages and drawbacks w.r.t. the $\varepsilon$-constrained method.

## 4   Experiments

Our simple method can find broad applications in various deep learning tasks. We show that 1) the constrained and lexico optimization approach provides an efficient approach to profile non-convex Pareto sets, as we demonstrate in a fairness classification and a sparse representation learning task; and 2) lexico optimization alone offers a parameter-free and stable approach to incorporate auxiliary regularization/loss without hurting the optimization of the main loss function, as we show in a variety of tasks: fine-grained image classification, semi-supervised learning, and semantic segmentation.

**Hyperparameters and Settings**   We use Algorithm 1 with $\phi$ in (8). We always set $\alpha = \beta = 1$ except ablation studies and hence introduce no extra hyper-parameters beyond regular (stochastic) gradient descent. For step size, we fix the learning rate to be $10^{-2}$ when profiling Pareto sets, and use the default step size scheduler of the baseline methods of each of the other applications. All the results and error bars are estimated over 5 random trails.

**Sparsity Representation Learning**   We learn sparse feature representations on a supervised dataset $\mathcal{D}$ of $(x, y)$ pairs by applying a non-convex $L_p$ regularization:

$$f(\theta) = \mathbb{E}_{\mathcal{D}}[\ell(y, \phi_\theta(h_\theta(x)))], \qquad\qquad g(\theta) = \mathbb{E}_{\mathcal{D}}[\|h_\theta(x)\|_p^p],$$

where $\ell(\cdot, \cdot)$ is the data loss, $h_\theta(x) \mapsto z \in \mathbb{R}^m$ is a hidden feature map, $\phi_\theta$ is a prediction head, and $p$ is a power coefficient. We choose $p = 0.5$ which yields a non-convex sparsity penalty. We choose $h_\theta(x)$ to be a simple linear model on a simulated dataset (Figure 1(a)) and the pretrained BERT-base model (Devlin et al., 2018) on the IMDb binary classification dataset (Figure 1(b)) and on the 20 news group (NG20) dataset (Figure 1 (c)). NG20 is processed into 4-class classification following Han et al. (2016). For IMDb, we randomly sample 2K examples from the full training dataset.

We profile the Pareto set by first finding the two endpoints with lexico optimization (3) and its mirror version. Then we solve the constrained optimization (1) with $c$ distributed evenly between the two end points (see Section 2). Figure 1 shows that our method (blue points) profiles the Pareto fronts evenly, which are all non-convex. The red stars are obtained by gradient descent on the linear combination (2), with $\omega := \lambda/(1 + \lambda)$ taken from the uniform grid of $[0, 1]$, which can not profile the whole Pareto sets and yield unevenly distributed points as theory predicts.

**Fairness Regularization**   Standard ML models may yield unfair prediction in real world systems, making it necessary to regularize the training with fairness metrics. Assume we learn a predictor $\hat{y}_\theta(x)$ on a dataset $\mathcal{D}$ of $(x, y, a)$ pairs, where $x, y$ are feature and label, and $a$ a protected demographic attribute (e.g., male vs. female). We consider fairness-regularized learning with

$$f(\theta) = (\mathrm{cov}_{\mathcal{D}}(\hat{y}_\theta(x), a))^2, \qquad\qquad g(\theta) = \mathbb{E}_{\mathcal{D}}[\ell(y, \hat{y}_\theta(x))],$$

where $f$ measures fairness using the co-variance between the protected attribute $a$ and the prediction $\hat{y}_\theta(x)$; it can be viewed as a continuous surrogate of the *disparate impact* criterion (Calders & Verwer, 2010). $g$ is the standard classification loss. For the experiments, we use the *Adult Income*

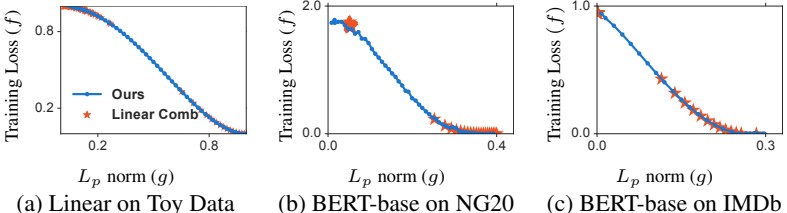

(a) Linear on Toy Data   (b) BERT-base on NG20   (c) BERT-base on IMDb

Figure 1: Our method (blue dots) profiles the Pareto fronts evenly in sparse representation learning, while linear combination (red stars) only finds parts of the Pareto front and yields unevenly distributed points.

dataset (Kohavi, 1996) for predicting whether the annual income of a person is $\geq \$50,000$, with gender as the protected attribute. Following the setting in related works (e.g. Martinez et al., 2020; Liu & Vicente, 2020), we randomly sample a subset of 5,000 data points from the original training set as our training set.

Figure 2 (a,left) shows the Pareto front profiled by our method (blue dots) and the linear combination loss (2) (red dots) with $\lambda/(1+\lambda)$ uniformly taken from a $20 \times 1$ grid of $[0, 1]$. Our method finds a good non-convex Pareto set, while (interestingly) the linear combination method finds a different convex, but sub-optimal Pareto set, in a different local mode of the landscape. Note that since our Pareto front is concave, the points on it are the local maxima (not minima) of the linear combination loss (2), and hence can not be found by gradient descent on (2) even if the corresponding $\lambda$ is given. Figure 2(a,right) shows the prediction accuracy on the true disparate impact score on the testing data; Figure 2(b) shows the trajectory of $f$ and $g$ in Algorithm 1 for the Lexico case (3), and the gradient descent of (2) with a large $\lambda$.

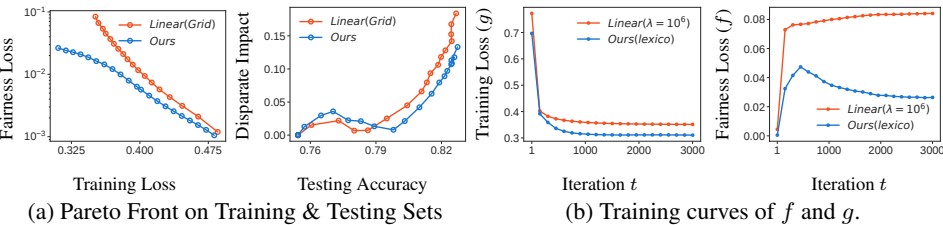

(a) Pareto Front on Training & Testing Sets   (b) Training curves of $f$ and $g$.

Figure 2: Fairness regularization on the Adult Income Dataset.

**Lexicographic $\ell_2$ Regularization**   Lexico optimization (3) provides a parameter-free approach to adding secondary regularization without hurting the optimization of the main loss. We apply it to fine-tune ImageNet pretrained models on fine-grained image classification with $L_2$ regularization:

$$f(\theta) = \|\theta\|_2^2, \qquad\qquad g(\theta) = \mathbb{E}_{\mathcal{D}}[\ell(y, \hat{y}_\theta(x))],$$

where the main loss $g$ is the typical training loss, and the secondary loss $f$ is the squared $L_2$ norm. Recent works (e.g. Huang et al., 2018; Salman et al., 2020) have shown that the $L_2$ regularization (when combined linearly as (2)) helps improve the accuracy with $\lambda$ optimally selected with a holdout data (which causes high training cost). Lexico optimization eliminates the need for tuning $\lambda$.

For our experiments, we use two fine-grained classification datasets: Oxford Flower (Nilsback & Zisserman, 2008) and Stanford Car (Krause et al., 2013), with two deep neural architectures, including EfficientNet (Tan & Le, 2019), and AlphaNet (Wang et al., 2021); AlphaNet is the state-of-the-art mobile model on ImageNet. We use the official checkpoints provided by (Tan & Le, 2019) and (Wang et al., 2021). Table 1 shows that the Lexico approach, which requires no tuning of $\lambda$, is almost on par with the best results of the linear combination approach with the best $\lambda$ on a grid. The grid search space is given by the baseline code of (Huang et al., 2018; Salman et al., 2020).

**Lexico Optimization for Semi-Supervised Learning**   Recently, various unsupervised auxiliary losses based on data augmentation have been used in semi-supervised learning (Bachman et al., 2019; Berthelot et al., 2019; Gong et al., 2020; Sohn et al., 2020; Xie et al., 2019, 2020). Typically, the auxiliary loss is linearly combined with the main supervised loss. Instead of this approach, we combine these two using lexico optimization. Following (e.g. Xie et al., 2019), we consider

$$f(\theta) = \mathbb{E}_{T \sim \mathcal{T}} \mathbb{E}_{x \sim \mathcal{D}_u}[\text{KL}(\hat{y}_\theta(x) \,||\, \hat{y}_\theta(T(x)))], \qquad g(\theta) = \mathbb{E}_{(x,y) \sim \mathcal{D}_s}[\ell(\hat{y}_\theta(x), y)],$$

| | | $\ell_2$ regularization (linear combination) | | | | | Lexico |
|---|---|---|---|---|---|---|---|
| | | $\infty$ | $2.5 \times 10^7$ | $2.5 \times 10^6$ | $2.5 \times 10^5$ | $2.5 \times 10^4$ | |
| Oxford Flower | AlphaNet-A0 | 96.6±0.2 | 96.8±0.2 | 97.2±0.1 | **97.7±0.2** | 97.3±0.1 | **97.6±0.1** |
| | EffNet-B0 | 96.4±0.1 | 96.7±0.1 | 96.9±0.1 | **97.2±0.1** | 96.9±0.1 | **97.1±0.1** |
| Stanford Car | AlphaNet-A0 | **91.6±0.1** | 91.3±0.2 | 91.0±0.1 | 90.8±0.2 | 90.6±0.2 | **91.6±0.1** |
| | EffNet-B0 | **91.0±0.1** | 90.7±0.2 | 90.6±0.2 | 90.6±0.1 | 90.3±0.1 | **91.0±0.1** |

Table 1: Top-1 test accuracy on fine-grained image classification tasks obtained by lexicographic optimization, and the linear combination with different $\lambda$.

where the main loss $g$ is the typical training loss on a labeled dataset $\mathcal{D}_s$ of $(x, y)$ pairs, and the secondary loss $f$ is the unsupervised data augmentation (UDA) loss (e.g. Xie et al., 2019) on an unlabeled dataset $\mathcal{D}_u$ consisting of $x$ only; intuitively, $f$ measures the consistency of the prediction from $x$ and a perturbed $x' = \mathrm{T}(x)$ using a randomly transform $\mathrm{T}$ (e.g., rotation, flipping, etc.). Table 2 shows that the result of lexico optimization is comparable with or even better than the baseline provided by e.g., Xie et al. (2019), which uses linear combination with $\lambda$ gradually decayed during training from a very large value (e.g. $+\infty$) to 0.2. The improvement of the lexico method is especially significant when the size of the labelled dataset is small (e.g. 100).

| Method | CIFAR-10 | | | CIFAR-100 | | |
|---|---|---|---|---|---|---|
| | 100 labels | 250 labels | 1000 labels | 1000 labels | 2500 labels | 10000 labels |
| UDA | 75.28±2.39 | 91.18±1.08 | **94.56±0.21** | 54.85±0.83 | 66.37±0.28 | **67.42±0.22** |
| UDA + Lexico | **77.61±1.83** | **91.86±0.85** | 94.53±0.15 | **56.91±0.78** | 67.06±0.26 | 67.57±0.14 |

Table 2: Testing accuracy (%) of different methods on CIFAR-100 and CIFAR-10.

**Lexico Optimization for Semantic Segmentation**   Recent state-of-the-art (SOTA) semantic segmentation methods are trained with the linear combination of loss functions from two models with different complexity/resolutions (MMSegmentation, 2020):

$$f(\theta) = \mathbb{E}_{\mathcal{D}}[\ell(\tilde{y}_\theta(x), y)], \qquad g(\theta) = \mathbb{E}_{\mathcal{D}}[\ell(\hat{y}_\theta(x), y)],$$

where $\hat{y}_\theta$ is the main prediction model and $\tilde{y}_\theta$ an auxiliary model with lower complexity and cost. Table 3 shows that lexico optimization yields better or comparable results on these two losses than linear combination. Here we use Swin Transformer (Liu et al., 2021), a recent-proposed SOTA model, as our backbone and use the code provided by (MMSegmentation, 2020). We use the Cityscapes (Cordts et al., 2016) benchmark, and calculate the main loss with UperNet (Xiao et al., 2018), and use FCN (Long et al., 2015) on the 3/4 of the total depth of the network to calculate the auxiliary loss.

| Model | Linear Combination $\lambda$ | | | | | Lexico |
|---|---|---|---|---|---|---|
| | 10 | 5 | 10/3 | 2.5 | 2 | |
| Swin-Base UperNets | 79.3±0.3 | 79.6±0.2 | 79.5±0.1 | 79.5±0.1 | 79.1±0.3 | **79.7±0.1** |
| Swin-Base FCN | 78.3±0.2 | 78.2±0.1 | 78.1±0.2 | 78.2±0.2 | 78.0±0.2 | **78.5±0.1** |

Table 3: mIoU for segmentation on Cityscapes validation set. All the models are trained with 40K iterations and batch size 8. The results are averaged over five trials.

# 5   Conclusion, Limits, Future Directions

We provide a joint treatment of constrained and lexicographic optimization with a general dynamic barrier gradient descent algorithm, which provides an efficient tool for profiling non-convex Pareto sets and incorporating secondary losses as we show in experiments. Our work has a potential positive social impact for providing new tools for more efficient and fairness ML. Currently, our work focuses on a single constraint, which yields a particularly simple algorithm, and is sufficient for broad applications. The extension to multiple constraints is straightforward (see Appendix 7), but the calculation of $\lambda_t$ (which would be a vector) requires solving a convex quadratic program that can be expensive when the number of constraints is large. In practice, we recommend to reform the multiple constraint problems into a single constraint one and then apply our method. For theoretical analysis, our results focus on the continuous time limit and the exact (rather than mini-batch) gradient. A future direction is to relax these constraints.

**Acknowledgements** The work is conducted in the statistical learning and AI group in the computer science department at UT Austin, which is supported in part by CAREER-1846421, SenSE-2037267, EAGER-2041327, and Office of Navy Research, and NSF AI Institute for Foundations of Machine Learning (IFML). We would like to thank the anonymous reviewers and the area chair for their thoughtful comments and efforts towards improving our manuscript.

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
