# 6 Proofs

## 6.1 Proof of Theorem 3.2

*Proof of Theorem 3.2.* Recall that

$$v_t = \arg\min_{v \in \mathbb{R}^d} \left\{ \|\nabla f(\theta_t) - v\|^2 \quad s.t. \quad \nabla g(\theta_t)^\top v \geq \phi(\theta_t) \right\}, \tag{20}$$

and its solution is $v_t = \nabla f(\theta_t) + \lambda_t \nabla g(\theta_t)$, where $\lambda_t$ is the Lagrange multiplier of (20). We have $\nabla g(\theta_t)^\top v_t \geq \phi(\theta_t)$ by the constraint in (20). In addition, by the slack condition of (20), we have

$$\lambda_t \nabla g(\theta_t)^\top v_t = \lambda_t \phi(\theta_t),$$

which plays an important role in the derivation below.

If $g(\theta_t) - c > 0$, we have $P_\mu(\theta_t) = f(\theta_t) + \mu(g(\theta_t) - c)$, and hence

$$\begin{aligned}
\frac{d}{dt} P_\mu(\theta_t) &= -(\nabla f(\theta_t) + \mu \nabla g(\theta_t))^\top v_t \\
&= -\|v_t\|^2 - \mu \nabla g(\theta_t)^\top v_t + \lambda_t \nabla g(\theta_t)^\top v_t \quad \text{//using } \nabla f(\theta_t) = v_t - \lambda_t \nabla g(\theta_t) \\
&\leq -\|v_t\|^2 - (\mu - \lambda_t)\phi(\theta_t) \quad \text{//using } \mu \nabla g(\theta_t)^\top v_t \geq \mu\phi(\theta_t) \text{ and } \lambda_t \nabla g(\theta_t)^\top v_t = \lambda_t\phi(\theta_t) \\
&= -\|v_t\|^2 - (\mu - \lambda_t)[\phi(\theta_t)]_+. \quad \text{//}\phi(\theta_t) \geq 0 \text{ by sign condition (7)}
\end{aligned}$$

If $g(\theta_t) - c < 0$, we have $P_\mu(\theta_t) = f(\theta_t)$ and hence $\frac{d}{dt} P_\mu(\theta_t) = -\nabla f(\theta_t)^\top v_t$.

If $g(\theta_t) - c = 0$, we are on the non-differentiable points of $P_\mu(\theta_t) = f(\theta_t) + \mu[g(\theta_t) - c]_+$. In this case, because the moving direction is orthogonal to $\nabla g(\theta_t)$, that is, $\frac{d}{dt} g(\theta_t) = \nabla g(\theta_t)^\top v_t = 0$, we have $\frac{d}{dt}[g(\theta_t) - c]_+ = 0$. Therefore, we also have $\frac{d}{dt} P_\mu(\theta_t) = -\nabla f(\theta_t)^\top v_t$.

Therefore, when $g(\theta_t) - c \leq 0$, we have

$$\begin{aligned}
\frac{d}{dt} P_\mu(\theta_t) &= -\nabla f(\theta_t)^\top v_t \\
&= -\|v_t\|^2 + \lambda_t \nabla g(\theta_t)^\top v_t \quad \text{//using } \nabla f(\theta_t) = v_t - \lambda_t \nabla g(\theta_t) \\
&= -\|v_t\|^2 + \lambda_t \phi(\theta_t) \quad \text{//by the slack condition } \lambda_t \nabla g(\theta_t)^\top v_t = \lambda_t \phi(\theta_t) \\
&= -\|v_t\|^2 - \lambda_t[-\phi(\theta_t)]_+. \quad \text{//}\phi(\theta_t) \leq 0 \text{ by sign condition (7)}
\end{aligned}$$

Combining the cases above, we have

$$\begin{aligned}
\frac{d}{dt} P_\mu(\theta_t) &\leq -\|v_t\|^2 - (\mu - \lambda_t)[\phi(\theta_t)]_+ - \lambda_t[-\phi(\theta_t)]_+ \\
&= -K_{\mu - \lambda_t}(\theta_t, \lambda_t). \quad \text{//by the definition of the KKT score in (9)} \tag{21}
\end{aligned}$$

$\square$

## 6.2 Proof of Corollary 3.3

*Proof of Corollary 3.3.* i) At each time point $t \in [0, \infty)$, dividing both sides of (11) by $\mu > 0$ and taking $\mu \to +\infty$ gives

$$\frac{d}{dt}[g(\theta_t) - c]_+ \leq -[\phi(\theta_t)]_+ \leq 0.$$

Integrating this on time interval $[0, t]$ gives

$$\min_{s \in [0,t]}[\phi(\theta_s)]_+ \leq \frac{1}{t}\int_0^t [\phi(\theta_s)]_+ ds \leq \frac{1}{t}([g(\theta_0) - c]_+ - [g(\theta_t) - c]_+) \leq \frac{1}{t}[g(\theta_0) - c]_+.$$

ii) Taking $\mu = 0$ in (11), we obtain

$$\frac{d}{dt} f(\theta_t) \leq -\|\nabla f(\theta_t) + \lambda_t g(\theta_t)\|^2 - \lambda_t[-\phi(\theta_t)]_+ + \lambda_t[\phi(\theta_t)]_+, \quad \forall t \in [0, \infty).$$

If we have $g(\theta_t) \leq c$ at time $t$, we have $[\phi(\theta_t)]_+ = [g(\theta_t) - c]_+ = 0$ by the sign condition (7), which yields (13). $\square$

## 6.3 Proof of Corollary 3.5

*Proof of Corollary 3.5.* 1) Under the sign condition of $\phi$ in (7), Assumption 3.4 is equivalent to say that $\lambda_t[\phi(\theta_t)]_+ \leq \lambda_{\max}[\phi(\theta_t)]_+$ for all time $t \in [0, +\infty)$. Therefore, following the definition of the KKT score in (9), we have $K_{\mu-\lambda_t}(\theta_t, \lambda_t) \geq K_{\mu-\lambda_{\max}}(\theta_t, \lambda_t)$. Hence, (14) holds following (11).

2) Take $\nu = \mu - \lambda_{\max}$ and integrate both sides of (14) in time interval $[0, t]$. We get

$$
\min_{s \in [0,t]} K_\nu(\theta_s, \lambda_s) \leq \frac{1}{t} \int_0^t K_\nu(\theta_s, \lambda_s) ds
$$

$$
\leq \frac{1}{t}(P_{\nu+\lambda_{\max}}(\theta_0) - P_{\nu+\lambda_{\max}}(\theta_t))
$$

$$
\leq \frac{1}{t}(P_{\nu+\lambda_{\max}}(\theta_0) - f^*),
$$

where we used that $P_{\nu+\lambda_{\max}}(\theta_t) = f(\theta_t) + (\nu + \lambda_{\max})[g(\theta_t) - c]_+ \geq f(\theta_t) \geq f^*$.

3) If $\theta_t$ is a fixed point (i.e., $d\theta_t/dt = -v_t = 0$), then $\frac{d}{dt}P_\mu(\theta_t) = 0$ for $\forall \mu \geq 0$. But $\frac{d}{dt}P_\mu(\theta_t) \leq -K_{\mu-\lambda_t}(\theta_t, \lambda_t)$ following (11). Hence $K_{\mu-\lambda_t}(\theta_t, \lambda_t) \leq 0$, for all $\mu \geq 0$. Because $\lambda_t < \infty$, taking $\mu = \lambda_t + \nu > \lambda_t$ gives that $K_\nu(\theta_t, \lambda_t) = 0$ for any $\nu \geq 0$. Hence the KKT condition holds. $\square$

## 6.4 Proof of Proposition 3.6

*Proof of Proposition 3.6.* 1) It is easy to note that

$$
\{\theta \colon g(\theta) \leq c_0\} = \{\theta \colon [g(\theta) - c]_+ \leq [g(\theta_0) - c]_+\}.
$$

Therefore, $\theta_t$ is contained in $\{\theta \colon g(\theta) \leq c_0\}$ because $[g(\theta_t) - c]_+ \leq [g(\theta_0) - c]_+$ for $\forall t \in [0, \infty)$ as shown in Corollary 3.3.

2) Note that

$$
\lambda_t = \max\left(\frac{\phi(\theta_t) - \nabla f(\theta_t)^\top \nabla g(\theta_t)}{\|\nabla g(\theta_t)\|^2}, \ 0\right) \leq \frac{\phi(\theta_t)}{\|\nabla g(\theta_t)\|^2} + \frac{\|\nabla f(\theta_t)\|}{\|\nabla g(\theta_t)\|}. \tag{22}
$$

Therefore, if $\max(\phi(\theta), \|\nabla f(\theta)\|, 1/\|\nabla g(\theta)\|) \leq M < +\infty$, it is easy to see that $\lambda_{\max} \leq M^3 + M^2 < \infty$.

Obviously, the condition that $\max(\phi(\theta), \|\nabla f(\theta)\|, 1/\|\nabla g(\theta)\|) < +\infty$ can be replaced by a finite bound of the right hand side of (22), which is a weaker condition. $\square$

## 6.5 Proof of Proposition 3.7

*Proof of Proposition 3.7.* In the lexico case, we have $c = g^*$ and hence $g(\theta_t) - c = g(\theta_t) - g^* \geq 0$ and $\phi(\theta_t) \geq 0$ for $\forall t \in [0, \infty)$. Plugging this into (11), we have for any $\mu \geq 0$,

$$
\frac{d}{dt}(f(\theta_t) + \mu(g(\theta_t) - g^*)) \leq -\|\nabla f(\theta_t) + \lambda_t \nabla g(\theta_t)\|^2 - (\mu - \lambda_t)\phi(\theta_t), \ \ \forall t \in [0, \infty).
$$

Taking integration on both sides on time interval $[0, t]$ gives:

$$
\int_0^t \left(\|\nabla f(\theta_s) + \lambda_s \nabla g(\theta_s)\|^2 + (\mu - \lambda_s)\phi(\theta_s)\right) ds \leq (f(\theta_0) - f(\theta_t)) + \mu(g(\theta_0) - g(\theta_t))) \tag{23}
$$

$$
\leq (f(\theta_0) - f^*) + \mu(g(\theta_0) - g^*)). \tag{24}
$$

Taking $\mu \to +\infty$ in (23) gives

$$
\int_0^t \phi(\theta_s) ds \leq g(\theta_0) - g^*, \tag{25}
$$

which gives $\min_{s \in [0,t]} \phi(\theta_s) \leq \frac{1}{t} \int_0^t \phi(\theta_s) ds \leq \frac{1}{t}(g(\theta_0) - g^*)$.

Taking $\mu = 0$ in (23) gives
$$\int_0^t \left\| \nabla f(\theta_s) + \lambda_s \nabla g(\theta_s) \right\|^2 ds \leq \int_0^t \lambda_s \phi(\theta_s) ds + (f(\theta_0) - f^*).$$
To get the desirable upper bound of $\int_0^t \left\| \nabla f(\theta_s) + \lambda_s \nabla g(\theta_s) \right\|^2 ds$, the main challenge is to bound $\int_0^t \lambda_s \phi(\theta_s) ds$. Because we assume $\phi(\theta_t) \leq \beta \left\| \nabla g(\theta_t) \right\|^\tau$ with $\tau \geq 1$, from Lemma 6.1, we have
$$\lambda_t \phi(\theta_t) \leq \left( \beta \left\| \nabla g(\theta_t) \right\|^{\tau-1} + \left\| \nabla f(\theta_t) \right\| \right) \beta^{\frac{1}{\tau}} \phi(\theta_t)^{1-\frac{1}{\tau}} \leq C_0 \phi(\theta_t)^{1-\frac{1}{\tau}},$$
where $C_0 = \sup_{\theta \in \mathbb{R}^d} \left( \beta \left\| \nabla g(\theta) \right\|^{\tau-1} + \left\| \nabla f(\theta) \right\| \right) \beta^{\frac{1}{\tau}}$. We have
$$\int_0^t \lambda_s \phi(\theta_s) ds \leq C_0 \int_0^t \phi(\theta_s)^{1-\frac{1}{\tau}} ds$$
$$\leq C_0 \left( \int_0^t \phi(\theta_s) ds \right)^{1-\frac{1}{\tau}} \left( \int_0^t 1 ds \right)^{\frac{1}{\tau}}$$
$$\leq C_0 \left( \int_0^t \phi(\theta_s) ds \right)^{1-\frac{1}{\tau}} t^{\frac{1}{\tau}}$$
$$\leq C_0 (g(\theta_0) - g^*)^{1-\frac{1}{\tau}} t^{\frac{1}{\tau}}, \qquad \textcolor{magenta}{\text{//using (25)}}$$
Therefore,
$$\min_{s \in [0,t]} \left\| \nabla f(\theta_s) + \lambda_t \nabla g(\theta_s) \right\|^2 \leq \frac{1}{t} \int_0^t \left\| \nabla f(\theta_s) + \lambda_t \nabla g(\theta_s) \right\|^2 ds$$
$$\leq \frac{1}{t} \int_0^t \lambda_s \phi(\theta_s) dt + \frac{1}{t} (f(\theta_0) - f^*)$$
$$\leq \frac{1}{t^{1-\frac{1}{\tau}}} C_0 \left( g(x_0) - g^* \right)^{1-\frac{1}{\tau}} + \frac{1}{t} (f(\theta_0) - f^*).$$
This completes the proof. $\qquad \square$

**Lemma 6.1.** *Assume*
$$\lambda_t = \max \left( \frac{\phi(\theta_t) - \nabla f(\theta_t)^\top \nabla g(\theta_t)}{\left\| \nabla g(\theta_t) \right\|^2}, \ 0 \right), \qquad \text{and} \qquad \phi(\theta_t) \leq \beta \left\| \nabla g(\theta_t) \right\|^\tau,$$
*where $\beta \geq 0$ and $\tau \geq 0$. Then we have*
$$\lambda_t \phi(\theta_t) \leq \left( \beta \left\| \nabla g(\theta_t) \right\|^{\tau-1} + \left\| \nabla f(\theta_t) \right\| \right) \beta^{\frac{1}{\tau}} \phi(\theta_t)^{1-\frac{1}{\tau}}.$$

*Proof.* From $\phi(\theta_t) \leq \beta \left\| \nabla g_t \right\|^\tau$, we have
$$\frac{\phi(\theta_t)}{\left\| \nabla g_t \right\|} \leq \beta \left\| \nabla g(\theta_t) \right\|^{\tau-1} \qquad \text{and} \qquad \frac{\phi(\theta_t)}{\left\| \nabla g_t \right\|} \leq \beta^{\frac{1}{\tau}} \phi(\theta_t)^{1-\frac{1}{\tau}}.$$
Since $\phi(\theta_t) \geq 0$,
$$\lambda_t = \frac{\max(0, \phi(\theta_t) - \nabla f(\theta_t)^\top \nabla g_t)}{\left\| \nabla g_t \right\|^2} \leq \frac{\phi(\theta_t) + \left\| \nabla f(\theta_t) \right\| \left\| \nabla g_t \right\|}{\left\| \nabla g_t \right\|^2}.$$
Therefore,
$$\lambda_t \phi(\theta_t) \leq \frac{\phi(\theta_t)^2}{\left\| \nabla g_t \right\|^2} + \left\| \nabla f(\theta_t) \right\| \frac{\phi(\theta_t)}{\left\| \nabla g_t \right\|}$$
$$\leq \left( \beta \left\| \nabla g(\theta_t) \right\|^{\tau-1} + \left\| \nabla f(\theta_t) \right\| \right) \frac{\phi(\theta_t)}{\left\| \nabla g_t \right\|}$$
$$\leq \left( \beta \left\| \nabla g(\theta_t) \right\|^{\tau-1} + \left\| \nabla f(\theta_t) \right\| \right) \beta^{\frac{1}{\tau}} \phi(\theta_t)^{1-\frac{1}{\tau}}.$$

$\qquad \square$

## 6.6 Proof of Proposition 3.8

*Proof of Proposition 3.8.* Because $\lim_{t\to\infty}\theta_t = \theta^*$ and $\phi$, $\nabla g$ are continuous, we have $\lim_{t\to\infty}\phi(\theta_t) = \phi(\theta^*)$, and $\lim_{t\to\infty}\|\nabla g(\theta_t)\| = \|\nabla g(\theta^*)\|$.

Eq. (25) shows that $\int_0^\infty \phi(\theta_t)dt \leq g(\theta_0) - g^* < +\infty$. Therefore, we must have $\lim_{t\to\infty}\phi(\theta_t) = \phi(\theta^*) = 0$. On the other hand, we have $\lim_{t\to\infty}\|v_t\| = 0$ by Assumption. This yields $\lim_{t\to\infty}K_\nu(\theta_t,\lambda_t) = 0$ for any $\nu > 0$ and applying Proposition 6.2 yields the result. $\qquad\square$

**Proposition 6.2.** *Consider the lexico optimization* (3) *with* $c = g^* = \inf_\theta g(\theta) > -\infty$, *and* $\phi$ *satisfying the sign condition* (7). *Assume* $f$, $g$, $\nabla g$, $\phi$ *are continuously differentiable. Let* $\{(\theta_t,\lambda_t)\colon t \geq 0\}$ *be a sequence which satisfies* $\lim_{t\to\infty}K_\nu(\theta_t,\lambda_t) = 0$ *for some* $\nu > 0$. *Assume that* $\theta^*$ *is a limit point of* $\{\theta_t\}$ *as* $t \to \infty$, *and that the rank of the Hessian matrix* $\nabla^2 g(\theta)$ *equals a constant in a neighborhood of* $\theta^*$. *Then there exists* $\omega^* \in \mathbb{R}^d$, *such that the lexico KKT condition in* (16) *holds.*

*Proof.* In the lexico case, we have $\phi(\theta) \geq 0$. The assumption that $\lim_{t\to\infty}K_\nu(\theta_t,\lambda_t) = 0$ for some $\nu > 0$ yields say that $\lim_{t\to\infty}\phi(\theta_t) = 0$ and $\lim_{t\to\infty}\|\nabla f(\theta_t) + \lambda_t\nabla g(\theta_t)\| = 0$. We just need to prove that $\lim_{t\to\infty}\|\nabla g(\theta_t)\| = 0$ and use Proposition 6.3 below.

Because we assume that $\theta^*$ is a limit of $\{\theta_t\}$, there exists an increasing sequence $\{t_n\colon n = 1,2,\cdots\}$ such that $t_n \to +\infty$ and $\theta_{t_n} \to \theta^*$ as $n \to +\infty$. Because $\phi$, $\nabla g$ are continuous, we have $\lim_{n\to\infty}\phi(\theta_{t_n}) = \phi(\theta^*) = 0$, and $\lim_{n\to\infty}\|\nabla g(\theta_{t_n})\| = \|\nabla g(\theta^*)\|$.

Because $\phi(\theta^*) = 0$, by the sign condition of $\phi$, we have $\mathrm{sign}(g(\theta^*) - g^*) = \mathrm{sign}(\phi(\theta^*)) = 0$. Hence $g(\theta^*) = g^*$ and $\theta^*$ is a minimum point of $g$. This gives $0 = \|\nabla g(\theta^*)\| = \lim_{n\to\infty}\|\nabla g(\theta_{t_n})\|$. On the other hand, we have $\lim_{t\to\infty}\|\nabla f(\theta_t) + \lambda_t\nabla g(\theta_t)\| = 0$ by Assumption. Applying Proposition 6.3 yields the result. $\qquad\square$

**Proposition 6.3.** *Assume* $f$, $g$, $\nabla g$ *are continuously differentiable. Let* $\{(\theta_t,\lambda_t)\colon t \geq 0\}$ *be a sequence which satisfies* $\lim_{t\to\infty}\|\nabla g(\theta_t)\| = 0$ *and* $\lim_{t\to\infty}\|\nabla f(\theta_t) + \lambda_t\nabla g(\theta_t)\| = 0$. *Assume that* $\theta^*$ *is a limit point of* $\{\theta_t\}$ *as* $t \to \infty$, *and that the rank of the Hessian matrix* $\nabla^2 g(\theta)$ *equals a constant in a neighborhood of* $\theta^*$. *Then there exists* $\omega^* \in \mathbb{R}^d$, *such that the lexico KKT condition in* (16) *holds.*

*Proof.* Because we assume that $\theta^*$ is a limit of $\{\theta_t\}$, there exists an increasing sequence $\{t_n\colon n = 1,2,\cdots\}$ such that $t_n \to +\infty$ and $\theta_{t_n} \to \theta^*$ as $n \to +\infty$. Because $\nabla g$ is continuous, we have $\nabla g(\theta^*) = \lim_{n\to+\infty}\nabla g(\theta_{t_n}) = 0$.

$$
\begin{aligned}
\|\nabla f(\theta_t) + \lambda_t\nabla g(\theta_t)\| &= \|\nabla f(\theta_t) + \lambda_t(\nabla g(\theta_t) - \nabla g(\theta^*))\| \\
&= \left\|\nabla f(\theta_t) + \lambda_t\nabla^2 g(\theta_t')(\theta_t - \theta^*)\right\| \quad \text{//Taylor expansion} \\
&= \left\|\nabla f(\theta_t) + \nabla^2 g(\theta_t')\omega_t'\right\|,
\end{aligned}
$$

where $\theta_t'$ is a convex combination of $\theta_t$ and $\theta^*$, and we defined $\omega_t' = \lambda_t(\theta_t - \theta^*)$.

Define $\omega_t = (\nabla^2 g(\theta_t'))^+\nabla f(\theta_t)$, where $(\nabla^2 g(\theta_t'))^+$ denotes the Moore–Penrose pseudo-inverse of matrix $\nabla^2 g(\theta_t')$, which satisfies that

$$
\omega_t = \arg\min_{\omega\in\mathbb{R}^d}\left\{\|\omega\| \quad s.t. \quad \omega \in \arg\min_w \left\|\nabla f(\theta_t) + \nabla^2 g(\theta_t')w\right\|\right\}.
$$

Therefore,

$$
\left\|\nabla f(\theta_t) + \nabla^2 g(\theta_t')\omega_t\right\| \leq \left\|\nabla f(\theta_t) + \nabla^2 g(\theta_t')\omega_t'\right\| = \|\nabla f(\theta_t) + \lambda_t\nabla g(\theta_t)\|.
$$

Because $\|\nabla f(\theta_{t_n}) + \lambda_{t_n}\nabla g(\theta_{t_n})\| \to 0$ as $n \to +\infty$, we have $\left\|\nabla f(\theta_{t_n}) + \nabla^2 g(\theta_{t_n}')\omega_{t_n}\right\| \to 0$.

Note that $\theta_{t_n} \to \theta^*$ and $\theta_{t_n}' \to \theta^*$ as $n \to +\infty$. By the constant rank assumption and Corollary 3.5 of Stewart (1977) (rephrased in Lemma 6.4), we have that $(\nabla^2 g(\theta_{t_n}'))^+ \to (\nabla^2 g(\theta^*))^+$ and hence $\omega_{t_n} \to \omega^*$ as $n \to +\infty$, where $\omega^* := (\nabla^2 g(\theta^*))^+\nabla f(\theta^*)$.

Therefore, we have $\left\|\nabla f(\theta_t) + \nabla^2 g(\theta_t')\omega_t\right\| \to \left\|\nabla f(\theta^*) + \nabla^2 g(\theta^*)\omega^*\right\|$., which implies that $\left\|\nabla f(\theta^*) + \nabla^2 g(\theta^*)\omega^*\right\| = 0$. $\qquad\square$

**Lemma 6.4** (Corollary 3.5 of Stewart (1977))**.** *Let $\{A_t\}$ be a sequence of matrices that converges to $A_*$ as $t \to \infty$. Then a necessary and sufficient condition that*

$$\lim_{t \to \infty} A_t^+ = A_*^+$$

*is that $rank(A_t) = rank(A_*)$ for sufficiently large $t$.*

### 6.7 Primal-Dual Methods and Non-convexity

**Example 6.5** (Expansion of Example 3.9)**.** *Consider a simple problem of form*

$$\min_{\theta \in \mathbb{R}} |\theta - 1|^\alpha \quad s.t. \quad |\theta|^\alpha \le b^\alpha,$$

*where $b \in (0, 1)$ and $\alpha > 0$. Note that all $\alpha > 0$ yields an equivalent problem whose true solution and the associated Lagrange multiplier are*

$$\theta^* = b, \qquad\qquad\qquad \lambda^* = (1 - b)^{\alpha-1}/b^{\alpha-1}. \qquad\qquad (26)$$

*However, the choice of $\alpha$ changes the convexity and hence impacts the results of different algorithms. The dual function is*

$$\Phi(\lambda) = \min_\theta |\theta - 1|^\alpha + \lambda(|\theta|^\alpha - b^\alpha).$$

*If $\alpha \le 1$, we can show $\Phi(\lambda) = \min(1, \lambda) - \lambda b^\alpha$. Maximizing it yields $\hat{\lambda}^* = 1$, which is different from the true $\lambda^*$ in (26). In this case, minimizing $f(\theta) + \hat{\lambda}^* g(\theta)$ would yield $\hat{\theta}^* = 0$, or 1, which is different from the true solution $\theta^* = b$.*

*Even if we plugin the true Lagrange multiplier $\lambda^*$ in (26) and minimize $f(\theta) + \lambda^* g(\theta)$ (see the green curve in Figure (5)), we would still yield $\hat{\theta}^* = 0$ or 1. As shown in Figure 5, the true solution $\theta^* = b$ is a local maximum (rather than minimum) of $f(\theta) + \lambda^* g(\theta)$.*

*In comparison, Algorithm 1 can find the true solution for all $\alpha > 0$. This is because the algorithm jointly updates $\theta_t$ and $\lambda_t$, such that the update direction $v_t = \nabla f(\theta_t) + \lambda_t \nabla g(\theta_t)$ always points towards the true solution $\theta^*$ (see the red curve in Figure 5). The way we update $\lambda_t$ in Algorithm 1 can be viewed as a "closed loop controller" that adjust $\lambda_t$ in a way that stabilizes $\theta_t$ around $\theta^*$.*

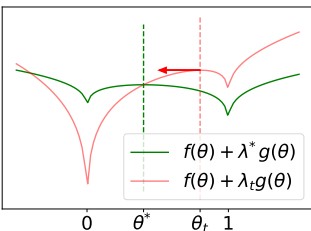

Figure 3: Green curve: $f(\theta) + \lambda^* g(\theta)$ with the true Lagrange multiplier $\lambda^*$ associated with the true solution $\theta^*$ (green dash) in (26); Red curve: $f(\theta) + \lambda_t g(\theta)$ with the $\lambda_t$ associated with $\theta_t$ (red dash) in Algorithm 1; the red arrow shows the update direction of in Algorithm 1, which points towards the true solution $\theta^*$. But minimizing $f(\theta) + \lambda^* g(\theta)$ would yield $\theta = 0$ or 1, which is not the true solution; instead, the true solution $\theta^*$ (green dash line) is a local maximum of $f(\theta) + \lambda^* g(\theta)$.

## 7 Multiple Constraints

Consider the constrained optimization with $m$ different constraints $\{g_i\}_{i=1}^m$:

$$\min_{\theta \in \mathbb{R}^d} f(\theta) \quad s.t. \quad g_i(\theta) \le c_i, \qquad \forall i \in [m].$$

To solve this problem, we iteratively update the parameter by

$$\theta_{t+1} \leftarrow \theta_t - \epsilon v_t,$$

where $v_t$ is choosen to be solve the following optimization:

$$v_t = \arg\min_{v \in \mathbb{R}^d} \left\{ \frac{1}{2} \|\nabla f(\theta_t) - v_t\|^2 \quad s.t. \quad \nabla g_i(\theta_t)^\top v \ge \phi_i(\theta_t) \right\}, \qquad (27)$$

where $\phi_i$ is a control function associated with constraint $g_i$, which should satisfy the sign condition:

$$\text{sign}(\phi_i(\theta)) = \text{sign}(g_i(\theta) - c_i), \quad \forall \theta \in \mathbb{R}^d, \quad \forall i \in [m].$$

The Lagrange dual of (27) is

$$\max_{\lambda \geq 0} \Phi(\lambda), \qquad \Phi(\lambda) = \min_{v \in \mathbb{R}^d} \left\{ \frac{1}{2} \|\nabla f(\theta_t) - v_t\|^2 - \sum_{i=1}^{m} \lambda_i (\nabla g_i(\theta_t)^\top v - \phi_i(\theta_t)) \right\}, \qquad (28)$$

where $\lambda \in \mathbb{R}^m$ is the Lagrange multiplier and $\Phi$ is the dual function. It is easy to see that the optimal $v_t$ in (28) is achieved by

$$v_t = \nabla f(\theta_t) + \sum_{i=1}^{m} \lambda_i \nabla g_i(\theta_t),$$

and plugging it into $\Phi$ yields

$$\Phi(\lambda) = -\frac{1}{2} \left\| \nabla f(\theta_t) + \sum_{i=1}^{m} \lambda_i \nabla g_i(\theta_t) \right\|^2 + \sum_{i=1}^{m} \lambda_i \phi_i(\theta_t) + \frac{1}{2} \|\nabla f(\theta_t)\|^2.$$

Therefore, the dual problem reduces to

$$\min_{\lambda \geq 0} \left\{ \frac{1}{2} \left\| \nabla f(\theta_t) + \sum_{i=1}^{m} \lambda_i \nabla g_i(\theta_t) \right\|^2 - \sum_{i=1}^{m} \lambda_i \phi_i(\theta_t) \right\}.$$

This is a convex quadratic programming and can be solved efficiently for small $m$.

# 8 Additional Materials on Experiments

## 8.1 2D Toy

We use an $\mathbb{R}^d$-toy example with $f(\theta) = \|\theta - \theta^*\|^2$ and $g(\theta) = \|a^\top \theta + b\|^2$, where $\theta, \theta^*, a \in \mathbb{R}^2$, $b \in \mathbb{R}$ are constants. Note that the optima set of $g$ is the line $a^\top \theta + b = 0$.

In Figure 4, we show the trajectory of the algorithm with different $\alpha, \beta$ for constrained optimization (4(a) and (b)) and lexicographic optimization (4(c)). We can see that the algorithm converges to the true solution in call cases. As $\alpha$ and $\beta$ increase, the algorithm tends to descent the constraint faster. We use constant step size by default. In Figure 4(d), we show an example of trajectories when we use two popular adaptive step size schemes, Adam and RMSprop.

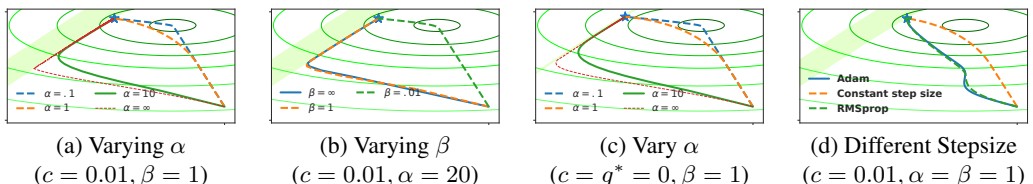

| (a) Varying $\alpha$ | (b) Varying $\beta$ | (c) Vary $\alpha$ | (d) Different Stepsize |
|---|---|---|---|
| $(c = 0.01, \beta = 1)$ | $(c = 0.01, \alpha = 20)$ | $(c = g^* = 0, \beta = 1)$ | $(c = 0.01, \alpha = \beta = 1)$ |

Figure 4: Algorithm trajectories on 2D toy with different threshold $c$ and $(\alpha, \beta)$ in (8). The contours denote the objective $f$ and the green areas are the feasible set. The stars are true solutions.

## 8.2 More Results

**Ablation on Batch Size**  For the fine-grained image classification tasks, we conduct an additional ablation study with different batch sizes. As shown in Table 4, different batch size yield similar performance.

## 8.3 Experiment Details

For the fairness regularization experiment, our model is a two-layer ReLU network with 50 hidden neurons. We use the SGD optimizer with a learning rate of 0.1 and a weight decay factor of $10^{-2}$.

| | | $\ell_2$ regularization (linear combination) | | | | | Lexico (batch size) | | |
|---|---|---|---|---|---|---|---|---|---|
| | | $\infty$ | $2.5 \times 10^7$ | $2.5 \times 10^6$ | $2.5 \times 10^5$ | $2.5 \times 10^4$ | 64 | 128 | 256 |
| Oxford Flower | AlphaNet-A0 | 96.6±0.2 | 96.8±0.2 | 97.2±0.1 | **97.7±0.2** | 97.3±0.1 | 97.6±0.1 | **97.7±0.1** | 97.6±0.1 |
| | EffNet-B0 | 96.4±0.1 | 96.7±0.1 | 96.9±0.1 | **97.2±0.1** | 96.9±0.1 | **97.1±0.1** | **97.1±0.1** | 97.0±0.1 |

Table 4: Top-1 test accuracy on fine-grained image classification tasks obtained by lexicographic optimization with different batch sizes, and the linear combination with different $\lambda$.

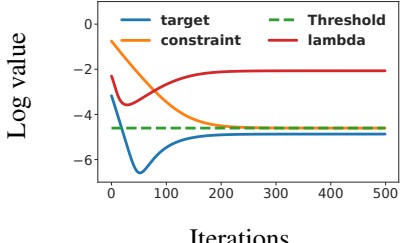

Figure 5: An example of the curve of constraint function, target function and $\lambda_t$ w.r.t. the iteration in our algorithm on the toy test function in 'sparsity representation learning' in Section 4.

We find the left-most endpoint of the Pareto set by solving the lexicographic optimization problem with Algorithm 1. The other points are found by solving the constrained optimization problem with uniformly increasing $c$, again using Algorithm 1. We set $\alpha = \beta = 1$ in Algorithm 1 for all the experiments.

For semi-supervised learning, we use WRN-28-10 as the network architecture, SGD optimizer with 0.03 learning rate, 0.9 momentum and $5 \times 10^{-4}$ weight decay. Each batch contains 512 unlabeled data and 64 labelled data. The learning rate is cosine decayed, and RandAugment is used as data augmentation to construct the consistency loss. We conduct experiments on 8 V100 GPUs.

For semantic segmentation, Cityscapes dataset labels 19 different categories (with an additional unknown class) and consists of 2975 training images, 500 validation images and 1525 testing images. We do an evaluation on the Cityscapes validation set in this paper. We closely follow the fine-tuning settings and hyper-parameters proposed in SWIN transformers (Liu et al., 2021). For the SWIN transformer, the window size is 7, the MLP ratio is 4, the drop path rate is 0.3 and the relative positional embedding is used. For the UperNet, the dropout rate is 0.1, and the number of channels is 512. During training, the image is re-scaled to $2049 \times 1025$ and a $769 \times 769$ random crop is applied. The batch size is 8 while synchronized batch normalization is used. During evaluation, the test image size is $2049 \times 1025$ and single scale evaluation with flipping is applied. The data process pipeline follows MMSegmentation (2020). We conduct experiments on 8 V100 GPUs.