# OpenReview forum: "Automatic and Harmless  Regularization  with Constrained and Lexicographic Optimization: A Dynamic Barrier Approach"
_NeurIPS.cc/2021/Conference — NeurIPS 2021 Poster_

### Official Review · Reviewer_Cn8E · 2021-07-11

**Rating:** 6
**Confidence:** 3

**Summary:**

This paper proposes a method called Dynamic Barrier Gradient Descent, which can solve constrained optimization problems, and more specific lexicographic optimization problems, in a way that is amenable for use in deep learning. The idea is to solve a convex quadratic program whose solution is an update direction that trades off minimization of the original loss and constraint satisfaction. The proposed method can be used in place (with no additional parameters) of the more conventional way of incorporating additional objectives (e.g. regularization), which is to simply optimize the linear combination with a tunable weighting coefficient. The authors prove convergence for the proposed method for general non-convex functions. They show empirically on a variety of deep learning tasks that their method is superior to the linear combination method in terms of profiling non-convex Pareto sets, and yields on par, or better final performance when used in place of the linear combination method.

**Limitations And Societal Impact:**

The authors have adequately addressed the limitations of their work.

**Main Review:**

The proposed method is novel, simple, and theoretically sound, and the paper is well written and easy to follow. The experiments include a variety of tasks, and support the claims of the paper. Although the absolute performance of using the proposed method might not be better than the linear combination method (sometimes better, mostly on par), the fact that the proposed method doesn’t require tuning makes this work significant.

Some comments:
- There are a couple of small typos: 1) line 146 “Assumption 3.1 ensures that is no local …” -> “Assumption 3.1 ensures that there is no local …” 2) line 180 “min and max yields a equivalent” -> “min and max yields an equivalent” 3) line 191 “maximizing f(\theta) + \lambda*g(\theta)” -> “minimizing f(\theta) + \lambda*g(\theta)” 4) line 248 “ We learning sparse feature…” -> “We learn sparse feature…” 5) line 266 misspelled fairness 6) Table 1 misspelled AlphaNet 7) line 321 “..which yields a particularly…” -> “which yields a particular..” 8) line 323 “...convex quadratic programming,” -> “convex quadratic program”
- What is L in Theorem 3.3 the second equation?
- Figure 1 (d), is the feasible region supposed to be a line since c = g* = 0?
- What are the \hat{g} used for all the experiments?
- Was there a reason why there are no l2 regularization experiments where you train from scratch? The lexicographic experiments are only for fine tuning.

Update:
It seems like there is an issue with the proof of lemma A.1 pointed out by reviewer TEma. As this affects the proof of theorem 3.3, I am going to lower my score. Once the authors address this issue and polish the paper with feedback from the reviews, I would be happy to champion this paper.


**Time Spent Reviewing:**

4

---

> ### Author Response · Authors · 2021-08-10
> **Reply to Reviewer Cn8E**
>
> Thanks a lot for the valuable feedback and
> for pointing out the typos. We will improve the draft based on your comments.
>
> Question: What is L in Theorem 3.3 the second equation?
>
> Answer: It a typo and should be $P$.
>
> Question: Figure 1 (d), is the feasible region supposed to be a line since c = g* = 0?
>
> Answer: This is a typo, the feasible set is c = 0.01.
>
> Question: What are the \hat{g} used for all the experiments?
>
> Answer: In our experiments, we set $\phi(\theta)$ to equation 6 and $\hat{g}$ is set to 0 since the loss is always positive in our experiments.
>
> Question: Was there a reason why there are no l2 regularization experiments where you train from scratch? The lexicographic experiments are only for finetuning.
>
> Answer: It is very expensive to train large models like BERT from scratch.
> Therefore, in practice,
> finetuning existing pre-trained models
> provides a much faster way for training models
> for different tasks,
> and it tends to yield good results
> because it transfers knowledge from the pre-trained models to different downstream tasks.
> In addition, because the downstream tasks have a relatively smaller number of data,
> finetuning comes to better results, and it is expensive to train large models like BERT from scratch. In addition,
> the results of finetuning are also more sensitive to the coefficient of L2 regularization,
> for which the benefit of our lexico method is more significant.

---

### Official Review · Reviewer_TEma · 2021-07-23

**Rating:** 5
**Confidence:** 4

**Summary:**

This work studies constrained and also lexicographic optimization problem. The authors first propose a general local descent algorithm which places a dynamic barrier constraint on the search direction. The authors then investigate the theoretical properties of their algorithm and claim that their algorithm is guaranteed to converge to a local optima for both constrained and lexico optimization with non-convex functions. Finally, the authors conduct experiments to evaluate the performance of their method on several real-world deep learning tasks.


**Limitations And Societal Impact:**

The authors have pointed out how to improve the paper such as extending their method to multiple constraints or SGD and discussed potential societal impact of their work.

**Main Review:**

The paper considers non-convex constrained and lexicographic optimization problem. This problem is important and has many applications in modern machine learning (especially in deep learning) tasks. The paper also provide empirical results of their method on some popular deep learning task such as fair classification and semantic segmentation. Their approach using a dynamic barrier function to guide the selection of update direction is interesting, despite that I have some concerns outlined below about their theoretical analysis, especially proof of the Theorems.

Cons:

(1) In the proof (Line 455-456) of Lemma A.1 in Appendix, the authors claim that "we must have $||\Delta g(\theta)||\geq e>0$ for some positive number e". This is not true as the minimizer of a continuous function on a closed set does not necessarily lies in the set. A simple counterexample is $g(\theta)=1/(1+e^{-\theta})$. Typically, for that claim to be true, $\bar{G}$ should be bounded. However this boundedness condition is not satisfied in the case of the paper.

(2) In the proof of Theorem 3.3 (Line 462), the authors use $-K_{\mu-\lambda_t}(\theta_t, \lambda_t)\leq -K_{\mu-\bar{\lambda}}(\theta_t, \lambda_t)$. Why the inequality holds? Because of $\bar{\lambda}\geq\lambda_t$? However $\bar{\lambda}\geq\lambda_t$ only when $g(\theta)\geq t$. When $g(\theta)\leq t$ during the update, we do not know which one is bigger. Moreover, the authors claim that $(\theta_t, \lambda_t)$ converges to a limit $(\theta^*, \lambda^∗)$ which satisfy the KKT condition just because $P_\mu(\theta_t)$ is decreasing function w.r.t $t$. This claim is not rigorous as the limit of the derivative of a bounded smooth monotone function is not always 0. The authors need additional assumption, i.e., the limit of the derivative exists.

(3) In the proof of Theorem 3.5, why (15) holds? According to my calculation, (15) is not correct and should be $\frac{d}{dt}f(\theta_t)=-\nabla f(\theta_t)^Tv_t\geq -||v_t||^2+\lambda_t \phi(\theta_t)$. The authors should carefully check for the correctness of the step. Moreover, in the last step of the proof (Line 481), the authors negligently treat the term $\int_0^T \phi(\theta_t)dt$ as a constant, which is incorrect as it depends on $T$. If assuming that $\phi(\cdot)$ is bounded, $\int_0^T \phi(\theta_t)dt$ should be $O(T^{1-\frac{1}{\tau}})$. Therefore, the final result should be $\frac{1}{T}\int_0^T||v_t||^2dt=O(1)$.

(4) There are many questionable claims. In lemma 3.4, the authors assume that the limit of $\theta_t$ exists, which may not be true in general. Moreover, in Theorem 3.5, the authors claim that the sequence $\{\theta_t\}$ converges to a limit $\theta^*$ and $\theta^*$ satisfies $\nabla g(\theta^*)=0$. Can the authors please explain this?

(5) The writing in the paper need to be polished, and there are many typos, i.e.,
- The authors interchange $x_t$ and $\theta_t$, $P_\mu$ and $L_\mu$ in the whole paper. It would be better if the symbols are consistent.
- In Line 50, ". But can not"->"and can not"
- In Line 79, "We should $\phi(\theta_t)$ to have"->"We let $\phi(\theta_t)$ have"
- In Theorem 3.3, "O(1/t)"->"O(1/T)"
- In Line 191, "maximizing"->"minimizing"


**Time Spent Reviewing:**

20

---

> ### Author Response · Authors · 2021-08-10
> **Reply to  Reviewer TEma**
>
> Thanks for the checking on the proofs and pointing out the typos.  We appreciate your comments. Your comments on Lemma A.1 and the existence of limits are correct; they involve some standard regularity conditions that we were not careful with, but do not impact the main results and are easily fixable (see below). Your other comments on the derivations are NOT correct, as we explain below.
>
> **Question: Condition of Lemma A.1.:**
> The issue you pointed out is correct. We will fix it by adding the condition that
> the sublevel  set $\\{\theta \colon g(\theta) \leq t_0 \\}$ for $t_0 = \max(c, (g(\theta_0))$ is bounded, which is a standard assumption to make (so we will add this in Assumption 3.1).  This fixes the problem because $\tilde g(\theta) := (g(\theta)-c)_+$ is a Lyapunov function of our method, and
> hence the trajectory is contained inside $\\{\theta\colon \tilde g(\theta) \leq  \tilde g(\theta_0)\\} = \\{\theta\colon g(\theta) \leq t_0\\}$, which is bounded and hence the $\bar{G}$ in Lemma A.1 can be replaced by a bounded set. Here $\tilde g(\theta)$ is a Lyapunov function because we monotonically decreases $g$ when $\theta_t\not\in \\{\theta\colon  g(\theta) \leq c\\}$  and then keep staying inside $\\{\theta\colon  g(\theta) \leq c\\}$ once $\theta_t$ enters it.
>
>
> **Question:Is $-K_{\mu-\lambda_t}(\theta_t, \lambda_t) \leq - K_{\mu-\bar \lambda}(\theta_t, \lambda_t)$ in Proof of Theorem 3.3 correct?**
> This step in our proof is correct.
> By the definition in Eq (7), we have $K_{\mu-\lambda_t}(\theta_t, \lambda_t) - K_{\mu-\bar \lambda}(\theta_t, \lambda_t) = (\bar \lambda - \lambda_t) \max(\phi(\theta_t), 0)$. If $g(\theta_t) \geq c$, we have $\bar \lambda \geq \lambda_t$.   On the other hand, if $g(\theta_t) <c$, we have $\max(\phi(\theta_t),0) = 0$ due to the sign condition that $\mathrm{sign}(\phi) = \mathrm{sign}(g-c)$. So the two cases combined give that $-K_{\mu-\lambda_t}(\theta_t, \lambda_t) \leq - K_{\mu-\bar \lambda}(\theta_t, \lambda_t)$.
>
>
>
> **"Question: Existence of Limits?"**
>  You are right that the existence of limits requires additional conditions. We spotted this problem when preparing the appendix.  So the versions of Theorem 3.3 and 3.5 in appendix did not contain claims of the limits.  For other places where we need to refer to a limit or fixed point (such as Lemma 3.4), we will add a phrase "if it exists". We thought it is better to avoid to claims on existence of limits since they involve generic regularity conditions that are not related to the special properties of our method.
>
>
> **"Question: Is Eq (15) in Theorem 3.5 Correct?"**
> Our Eq (15) is correct. This is because the slack condition of the optimization in Eq (4)  is $\lambda_t(\nabla g(\theta_t)\tt v_t - \phi(\theta_t)) = 0$, which gives $\lambda_t \nabla g(\theta_t)\tt v_t =\lambda_t \phi(\theta_t)$ (rather than $\lambda_t \nabla g(\theta_t)\tt v_t \geq \lambda_t \phi(\theta_t)$). We will add a remark on this. The same fact was used in the equation below Line 519,
> which had a remark.
>
> **"Question: Is the last step of proof of Theorem 3.5 (Line 481) correct?"**
> This step in our proof is correct. It is because by Eq (14), we have $\int_0^T \phi(\theta_t) dt \leq  g(\theta_0) - g^*$, where the right hand side  does not depend on  time $T$. We will add a remark.
>
> We will improve the draft based on your comments and check the proofs carefully.

---

> > ### Comment · Reviewer_TEma · 2021-09-02
> > **Thank you for the response**
> >
> > Thank the authors for the response which addresses my concerns about the proof of Theorem 3.3 and 3.5. I will according update my score. However, for my first question about the proof of Lemma A.1, the authors do not adequately address my concern as the bounded sublevel set assumption proposed by the authors is too strong which even does not hold for general convex function. Furthermore, I would suggest that the authors put more efforts on polishing the writing and check their proofs and claims carefully.

---

> > > ### Author Response · Authors · 2021-09-05
> > > **Thanks, and More on Condition of Lemma A.1**
> > >
> > >
> > > Dear reviewer,
> > > thanks for considering our response. We have been working on improving the paper and proofs based on your comments.
> > >
> > > We want to further clarify the condition Lemma A.1.
> > > Essentially, all we need here is to show that
> > > $$
> > > \sup_t \\{\lambda_t \colon ~~ g(\theta_t) \geq c\\} \leq \bar \lambda < +\infty.
> > > $$
> > > Meanwhile,
> > > we know that the trajectory $\{\theta_t\}$ satisfies $g(\theta_t) \leq c_0$, where $c= \max(c, g(\theta_0))$.
> > > Following the proof in Lemma A.1, a general sufficient condition to ensure the boundness condition of $\lambda_t$ is
> > > $$
> > > \lambda_t \leq \frac{\phi(\theta)}{||{\nabla g(\theta)}||^2} + \frac{||{\nabla f (\theta)}||}{||{\nabla g(\theta)}||} \leq \bar \lambda\colon ~~~~ \theta \in \Gamma,
> > > $$
> > > where $$\Gamma = \\{\theta \colon c\leq  g(\theta)\leq c_0\\}.$$
> > > This can be further simplified into the following condition
> > > $$
> > > \max(\phi(\theta), ||{\nabla f(\theta)}||) <+\infty, ~~~~
> > > ||{\nabla g(\theta)}|| > e > 0,~~~\forall \theta \in \Gamma,
> > > $$
> > > where $e$ is some positive number.
> > >
> > > The general conditions above do not require that $\Gamma$ is bounded,
> > > but if $\Gamma$ is indeed bounded and $\nabla g(\theta)$ is continuous, we can replace $||{\nabla g(\theta)}|| > e > 0$ with a slightly weaker condition  $||{\nabla g(\theta)}|| > 0$.

---

> ### Comment · Area_Chair_8BVf · 2021-08-22
> **Please read author response**
>
> Dear reviewer,
>
> You were the most critical about this paper. Can you read the author rebuttal and tell whether you stand by your score? Did the authors address your proof correctness concerns?
>
> Other reviewers: can you read this review and tell whether this changes your assessment?
>
> Thanks,
> the area chair

---

### Official Review · Reviewer_youY · 2021-07-30

**Rating:** 7
**Confidence:** 3

**Summary:**

In this paper, the authors introduce a unified first-order approach for non-convex constrained optimization with a single constraint, $\min_{\theta \in \mathbb{R}^d} f(\theta) \text{ s.t. } g(\theta) \le c
$, and simple bilevel programming (SBP), $\min_{\theta \in \mathbb{R}^d} f(\theta) \text{ s.t. } \theta \in \mathrm{argmin}_{\tilde{\theta} \in \mathbb{R}^d} g(\tilde{\theta})
$.

In a nutshell, the approach relies on the introduction of a *dynamic barrier* $\phi(\theta)$ whose sign coincides with that of the constraint in canonical form, and that is used to bound from below the rate of decrease of $g$. Precisely, the authors propose to use as update direction the vector $v_t$ closest in $L_2$ norm to $\nabla f$ among all directions whose inner product with $\nabla g$ is at least as large as $\phi(\theta)$. The resulting quadratic program can be solved in closed form, leading to $v_t = \nabla f(\theta_t) + \lambda_t \nabla g(\theta_t)$ where $\lambda_t$ can be obtained in closed form as well as the maximizer of the dual. In other words, the resulting algorithm can be informally understood as an adaptive variant of unconstrained minimization over a linear combination of $f$ and $g$, $\min_{\theta \in \mathbb{R}^d} f(\theta) + \lambda g(\theta)$, where the multiplier $\lambda$ is not fixed throughout but adaptively estimated for each $\theta$. Finally, the authors propose a unified choice for the dynamic barrier, $\phi(\theta) = \min(\alpha(g(\theta) - \hat{g}), \beta \vert\vert \nabla g(\theta) \vert\vert^{2})$, where $\hat{g}$ is either set to $c$ in the case of constrained minimization or a lower bound on $g^{*}$ for SBP, and $\alpha, \beta$ are hyperparameters that the authors suggest fixing to $1$ in practice.

The authors provide theoretical guarantees for both setups. For constrained optimization, it is shown that under certain regularity conditions, a continuous-time version of the algorithm with make the penalty function $P_{\mu}(\theta) = f(\theta) + \mu \max(g(\theta) - c, 0)$ decrease monotonically and, at convergence, the iterates $(\theta^{\*}, \lambda^{\*})$ will satisfy the KKT conditions for the problem, with “KKT suboptimality” decreasing at a linear rate. Analogously, the continuous-time version of the algorithm is shown to converge to a local minimum of $g$ with the resulting iterates $\theta_t$ being approximate minimizers of a sequence of constrained optimization problems with constraint $c_t := g(\theta_t)$. Moreover, these theoretical results are extended to the discrete-time setting in the Appendix.

Finally, the authors illustrate their approach on a suite of real-world tasks (sparse representation learning, fairness regularization, $L_2$ regularization, semi-supervised learning and semantic segmentation) as well as a toy 2D example. In these, they primarily compare their method against a linear combination baseline treating $\lambda$ as a (non-adaptive) hyperparameter with encouraging results.

**Limitations And Societal Impact:**

As stated in the main review, I believe the main limitations of the proposed approach are its inability to handle more than two losses and, perhaps somewhat less importantly, the lack of theoretical guarantees for stochastic optimization. Both of these limitations are clearly stated and discussed in the manuscript, which is appreciated.

Successfully extending the approach to an arbitrary number of losses with compelling empirical results on large scale multi-modal models could be a highly impactful contribution for the community.

**Main Review:**

## Assessment

From a pragmatic point of view, this manuscript contributes a gradient-based optimizer that allows practitioners typically relying on SGD-based optimization pipelines to incorporate either one (non-convex) constraint or a simple bilinear program (SBP) with an arguably low implementation burden and *de facto* no additional hyperparameters.

The proposed approach is, to the best of my knowledge, theoretically sound and assumptions and limitations are clearly stated throughout the manuscript.

Empirical results are shown for a wide range of real-world tasks suggesting that the proposed approach matches or outperforms unconstrained optimization over a linear combination of losses with a fixed, tunable weight.

The manuscript is also for the most part clearly written, precise, and plenty of intuition is given to the reader about the main ideas and concepts the proposed approach relies on. It would however benefit from proof-reading and additional polishing, as it contains abundant typos.

All things considered, I lean towards supporting acceptance of the manuscript in its current form. However, a few issues and limitations, some of which are discussed by the authors themselves, prevent me from further raising my score at this time, as I believe these limit potential impact to some extent. In brief:
+ The method is currently limited in practice to two losses, which undercuts its applicability to large multi-modal models and other related state-of-the-art approaches.
+ No theoretical guarantees are given for the stochastic setting, despite being arguably the most common use case for the proposed approach.
+ Relatedly, while experimental results were shown involving stochastic optimization, the effect of stochasticity on convergence rates and quality of the resulting solutions was not exhaustively studied empirically.



## Other comments / questions


1. There are some aspects of Example A.12 which I have not fully grasped. I would be grateful if the authors could provide additional detail on (1) how the red curve was computed; (2) how one can infer from that curve what the fixed point for the algorithm will be; and (3) to provide the value for all hyperparameters ($\alpha$, \$b$, $\phi$) to reproduce Figure 4.

    More generally, I am intrigued by how the discontinuity in $\nabla f$ at $\theta = 1$ for $\alpha < 1$ does not cause $\lambda_t$ to diverge as $\theta -> 1$ from the left.

2. During the abstract and introduction it is claimed that the method is optimal in the non-convex setting. However, what that optimality entails is not precisely stated until Section 3. It could help to give an informal description of these theoretical guarantees early on.

3. Is there a practical need to modify the computation of $\lambda_t$ in Algorithm 1 for numerical stability (e.g. by ensuring the denominator does not approach zero)?


## Typos (non-exhaustive)

Notation is at times inconsistent. For example, occasionally $\theta$ is changed to $x$, or $P_{\mu}$ is changed to $L_{\mu}$.

L7: emphasize -> emphasis

L15: leixco

L162: an local -> a local

Eq. 9: $g(\theta)$ -> $g(\theta) - c$

L199: $\theta^{\*} = 1 - b$ -> $\theta^{\*} = b$

L266: fairness

L269: predicted attribute -> protected attribute

L295: semi-surprised -> semi-supervised

Lemma A.1: formatting of supremum assumption

L455: close -> closed

Eq. below L461: $v(\theta_t)$ -> $v_t$

L546: multipler -> multiplier

**Time Spent Reviewing:**

10

---

> ### Author Response · Authors · 2021-08-10
> **Reply to Reviewer youY**
>
> We thank the reviewer for the valuable feedback and pointing out the typos. We will improve the draft based on your comments.
>
> **"Question: extension to the stochastic setting?"**
> We will defer a full course theoretical analysis for the stochastic setting to future works. Empirically, we have performed analysis on the dependency on batch size and found that our method is not very sensitive to the batch size in typical deep learning tasks. For example,  on the Oxford Flower fine-grained classification task, AlphaNet-A0 gets 97.6$\pm$0.1, 97.7$\pm$0.1 and 97.6$\pm$0.1 in accuracy for batch size 64, 128, and 256, respectively. For EfficientNet-B0, it gets 97.1$\pm$0.1, 97.1$\pm$0.1 and 97.0$\pm$0.1 for batch size 64, 128, and 256.
>
> **"Question: Regarding Example A.12"**
> Let $\theta_t$ be the solution at the $t$-th iteration and let $\lambda_t:=\max\left (\frac{\phi(\theta_t)-\nabla f(\theta_t)\tt \nabla g(\theta_t)}{||{\nabla g(\theta_t)}||^2}, 0 \right)$ be the corresponding Lagrange multiplier by our method as we write in the paper, then the red curve is the curve of $L_{\lambda_t}(\theta) = f(\theta) +\lambda_t g(\theta)$ as $\theta$ varies. In this way, we have $\theta_{t+1} = \theta_t + \epsilon v_t$, where  the update direction is $v_t =  -\nabla_{\theta} L_{\lambda_t}(\theta)\rvert_{\theta=\theta_t}$ (here $\lambda_t$ is viewed as a constant when taking gradient). The direction of $v_t$ is visualized by the red arrow in Figure 4, which  points towards the true solution  $\theta^*$, and hence indicates the our method converges towards  $\theta^*$. In comparison, the gradient of $L_{\lambda_*}(\theta) = f(\theta) + \lambda^* g(\theta)$, where $\lambda^*$ is the  Lagrange multiplier at the true solution $\theta^*$ at $\theta_t$, points away from the true solution (the green curve in Figure 4).
>
> The idea here is that, at each point $\theta= \theta_t$, we should choose $\lambda_t$ properly, such that the negative gradient of $L_{\lambda_t}(\theta)$ at $\theta=\theta_t$ points towards the true solution. One method provides one such rule for selecting $\lambda_t$, which ensures the convergence of the algorithm. Meanwhile, for non-convex problems, always setting $\lambda_t=\lambda^*$, or updating $\lambda_t$ by dual descent, may not have such a guarantee.
>
> To obtain a similar plot, try $\alpha = 0.5$, $b = 0.3$, $\theta_t = 0.7$, $\phi(\theta) = g(\theta)-c$ (which is equivalent to Eq 6 with $\alpha=\beta=1$ in this case). The actual values do not matter here.
>
> The discontinuity here is not a problem even though it is not covered by our current analysis (in fact, we found that the algorithm empirically works in a more challenging case when $f(x) = \log(|1-\theta|)$ and $g(\theta)-c=\log(|\theta|)-\log (b)$ whose  gradients diverge to infinite when $\theta\to 1$ or $0$).
>
> **"It could help to give an informal description of these theoretical guarantees early on."**
> Thanks for the suggestion of giving an informal description of theoretical guarantees earlier. We will incorporate it in the revision.
>
> **"Is there a practical need to stabilize $\lambda_t$?"** We did not find this to be a large issue. We found that the value of $\lambda_t$ can sometimes be very large while the algorithm continuous to work. But we think there should be some bad cases when stabilization is necessary; it is just that we did not encounter them in practice. One possible case is when $g$ has a zero gradient point outside of the feasible region (which was excluded by  Assumption 3.1), but in this case a very large  $\lambda_t$ is not necessarily a bad thing because it may help jump out of the local optima of $g$.
>
>
> **"Extension to multiple loss functions?"**
> The extension to multiple constraints is straightforward, except that $\lambda_t$ would be a vector and need to be calculated by solving a convex quadratic program (which would cast high computational cost when we have many constraints).  We will add a brief section in Appendix on the extension to multiple constraints. In the paper, we decided to focus on single constraints because it significantly simplified the presentation, and multiple constraints can be handled computationally efficiently by combining them into a single constraint (which may sacrifice some performance for speed).   In addition, it does not seem to be common to have multiple constraints in the lexico setting (which would correspond to having multiple main loss functions). Meanwhile, we do believe that further investigation of multiple constraints is a very important direction.

---

> > ### Comment · Reviewer_youY · 2021-08-31
> > **Response to rebuttal**
> >
> > I would like to thank the authors for their rebuttal. After going through the other reviews and their respective rebuttals, I lean towards maintaining my score and recommendation towards acceptance, pending any additional points brought up during the discussion.

---

> > > ### Author Response · Authors · 2021-09-05
> > > **Thanks!**
> > >
> > > Dear Reviewer,
> > >
> > > Thanks a lot for you consideration. We will further elaborate the points in your review in the revision of the draft.

---

### Decision · Program_Chairs · 2021-09-27

**Decision:**

Accept (Poster)

**Comment:**

This paper unfortunately only received three reviews.

One reviewer was negative due to concerns with some proofs but I think the authors have mostly addressed them.

Two reviewers were very positive and ready to back up the paper.

I would therefore like to accept the paper, assuming that the authors comply to the following two requests:

- Carefully proofread the paper and fix the numerous small typos

- Change the title, as the current title is awkward and uninformative (honestly with such a title, the authors are shooting themselves in the foot). I would suggest "Bi-objective optimization: a dynamic barrier approach" but other choices are of course possible.